# FRACTAL: State Space Model with Fractional Recurrent Architecture for Computational Temporal Analysis of Long Sequences

Mengqi Li [1]   Wensheng Lin [1 2]   Jinshuai Yang [1]   Lixin Li [1 2]

## Abstract

Effective sequence modeling fundamentally requires balancing the retention of unbounded history with the high-resolution detection of abrupt short-term variations common in real-world phenomena. However, existing state space models (SSMs) relying on high-order polynomial projection operators (HiPPO) face a critical trade-off where uniform measures dilute recent information to maintain timescale invariance, while exponential measures sacrifice global context to capture local dynamics. This paper proposes a Fractional Recurrent Architecture for Computational Temporal Analysis of Long sequences (FRACTAL), a novel architecture integrating fractional measure theory into recursive memory updates to address this limitation. By deriving projection operators with analytically characterized spectral properties and a tunable singularity index, the proposed method amplifies sensitivity to recent signal perturbations while preserving the spectral structure that encodes scale-invariant memory dynamics. This theoretical innovation is instantiated within a simplified diagonalized state space framework by modulating input projection initialization to enable simultaneous capture of multi-scale temporal features. FRACTAL achieves an average score of 87.11% on the Long Range Arena benchmark, including 61.85% on the ListOps task, outperforming the S5 model.

## 1. Introduction

Sequence modeling in continuous time requires resolving the fundamental tension between retention of unbounded historical context and high-resolution detection of recent input perturbations. While Long Short-Term Memory networks (Hochreiter & Schmidhuber, 1997) and Transformers (Vaswani et al., 2017) address these aspects through distinct mechanisms, state space models (SSMs) have emerged as a unified framework offering linear scaling complexity and rigorous grounding in continuous signal processing (Gu et al., 2022b; Tay et al., 2021). The theoretical foundation of modern SSMs lies in the high-order polynomial projection operators (HiPPO) framework (Gu et al., 2020), which formalizes memory as an online polynomial approximation problem governed by a probability measure. This measure serves as the primary mechanism determining the memory profile by dictating how the system weights historical information relative to the present.

Despite rapid architectural evolution from structured matrices (Gu et al., 2022b; Gupta et al., 2022) to selective state spaces (Gu & Dao, 2024), the foundational choice of measure remains largely unexplored. Existing HiPPO instantiations present a critical trilemma. The *Scaled Legendre* measure (LegS) assigns uniform weight to history, ensuring timescale invariance but progressively diluting the influence of recent inputs as time advances. The *Translated Laguerre* measure (LagT) prioritizes recency via exponential decay but imposes a fixed characteristic timescale that sacrifices robustness to time-warped signals. The *Translated Legendre* measure (LegT) (Voelker et al., 2019) provides high local resolution but catastrophically discards context beyond a fixed sliding window.

This theoretical gap limits the applicability of SSMs in physical and biological domains characterized by multi-scale dynamics. Real-world phenomena ranging from volatility clustering in financial markets (Mandelbrot, 2013) and $1/f$ noise in physiological signals (West, 2012), to the bursty nature of network traffic (Leland et al., 2002), exhibit long memory characteristics defined by power-law decay kernels. Such systems possess a heavy tail of history that interacts non-linearly with abrupt local variations. Standard SSMs governed by integer-order differential equations inherently

---

[1]School of Electronics and Information, Northwestern Polytechnical University, Xi'an, Shaanxi, China. [2]DecoreX Intelligent Technologies Co., Ltd., Xi'an 710075 China. Correspondence to: Lixin Li <lilixin@nwpu.edu.cn>, Wensheng Lin <linwest@nwpu.edu.cn>.

*Proceedings of the 43rd International Conference on Machine Learning*, Seoul, South Korea. PMLR 306, 2026. Copyright 2026 by the author(s).

impose exponential memory decay via LagT or uniform averaging via LegS, creating an impedance mismatch with these scale-free natural processes.

Bridging this gap motivates the adoption of fractional calculus, a mathematical framework generalizing differentiation to non-integer orders. Unlike integer-order operators that imply local dependencies, fractional operators naturally describe non-local memory and hereditary properties (Podlubny, 1998). A fractional-order measure provides a rigorous interpolation mechanism wherein adjusting the singularity index continuously tunes the memory kernel between a uniform distribution (pure history) and concentrated mass at the present (pure recency). This observation motivates FRACTAL (Fractional Recurrent Architecture for Computational Temporal Analysis of Long sequences), illustrated in Figure 1. The proposed approach generalizes the measure-theoretic foundation of SSMs by introducing a fractional-order measure with a tunable singularity index $\alpha \in [0, 1)$. In contrast to exponential decay, this measure induces a power-law weighting $(t - \tau)^{-\alpha}$ that creates a heavy tail for long-term retention while mathematically enforcing a singularity at the current timestep to amplify local sensitivity.

**Related Work.** The evolution of efficient sequence modeling has largely focused on computational tractability rather than measure-theoretic fundamentals. The Legendre Memory Unit (LMU) (Voelker et al., 2019) pioneered the use of orthogonal projections derived from delay differential equations but was restricted to fixed sliding windows. The S4 model (Gu et al., 2022b) extended this to infinite history using HiPPO-LegS matrices decomposed into Normal Plus Low-Rank (NPLR) structures, enabling efficient convolution via FFT. Subsequent simplifications, such as S4D (Gu et al., 2022a), DSS (Gupta et al., 2022), and S5 (Smith et al., 2023), demonstrated that the precise matrix structure is secondary to spectral properties, advocating for diagonalized Multi-Input Multi-Output (MIMO) systems initialized with HiPPO spectra. However, these works treat the HiPPO matrix as a static initialization artifact, ignoring the potential of the underlying measure to serve as a dynamic design parameter.

Parallel to the SSM development, fractional calculus has explored memory modeling, albeit often heuristically. Fractional gradient methods (Wang et al., 2017) introduced fractional-order derivatives into backpropagation to impose heavy-tailed memory on weight updates, though this approach focuses on optimization rather than sequence representation. In the domain of continuous dynamics, Physics-Informed Neural Networks have been rigorously extended to fractional orders (fPINNs) (Pang et al., 2019) to solve fractional differential equations. However, these approaches typically rely on expensive numerical solvers or automatic differentiation through fractional operators, rendering them

computationally prohibitive for large-scale sequence modeling tasks. The present work bridges these lineages by leveraging the optimal projection theory of HiPPO together with the memory properties of fractional calculus to derive an efficient state-space parameterization with an analytically characterized spectral structure.

The SSM landscape has recently bifurcated into two orthogonal paradigms. *Selective* (time-varying) SSMs, such as Mamba (Gu & Dao, 2024) and DeltaNet, achieve strong empirical performance by making the state-transition matrices input-dependent, effectively enabling data-dependent gating. These architectures are predominantly designed as generative language model backbones relying on large-scale pre-training and hardware-aware scan implementations. *Linear Time-Invariant* (LTI) SSMs (S4, S4D, DSS, S5, and this work) instead focus on the continuous-time mathematical measure $\mu^{(t)}$ used to compress history, and are evaluated by training from scratch to isolate the pure structural inductive bias of the sequence mixer. FRACTAL belongs strictly to the LTI category. Comparing a foundational, train-from-scratch LTI innovation against heavily optimised, pre-trained selective architectures conflates two distinct research paradigms and would obscure the specific theoretical contribution of the fractional measure; S5 therefore remains the appropriate rigorous baseline.

**Our Contributions.** This paper makes three contributions:

(1) **Theory.** We introduce a fractional-order power-law measure into the HiPPO projection framework as the first measure to simultaneously satisfy full-history retention, recency sensitivity, and scale invariance, by placing an integrable singularity at the current time while maintaining the polynomial decay over distant history that scale invariance structurally requires. Rigorous derivation of the resulting state-space dynamics establishes that Jacobi polynomials form the natural orthonormal basis, that the eigenvalues of the state transition matrix are invariant to the singularity index for all admissible parameter values, and that the input projection admits a complete closed-form solution.

(2) **Architecture.** The FRACTAL architecture operationalizes this theoretical framework within a multi-input multi-output structure amenable to parallel computation. A multi-channel configuration assigns heterogeneous memory parameters across state dimensions, enabling the simultaneous capture of patterns at multiple temporal scales.

(3) **Experiments.** Comprehensive experiments on the Long Range Arena benchmark validate the proposed approach. Results demonstrate state-of-the-art performance on tasks requiring long-range reasoning, validating the impact of fractional-order memory and multi-scale parameterization.

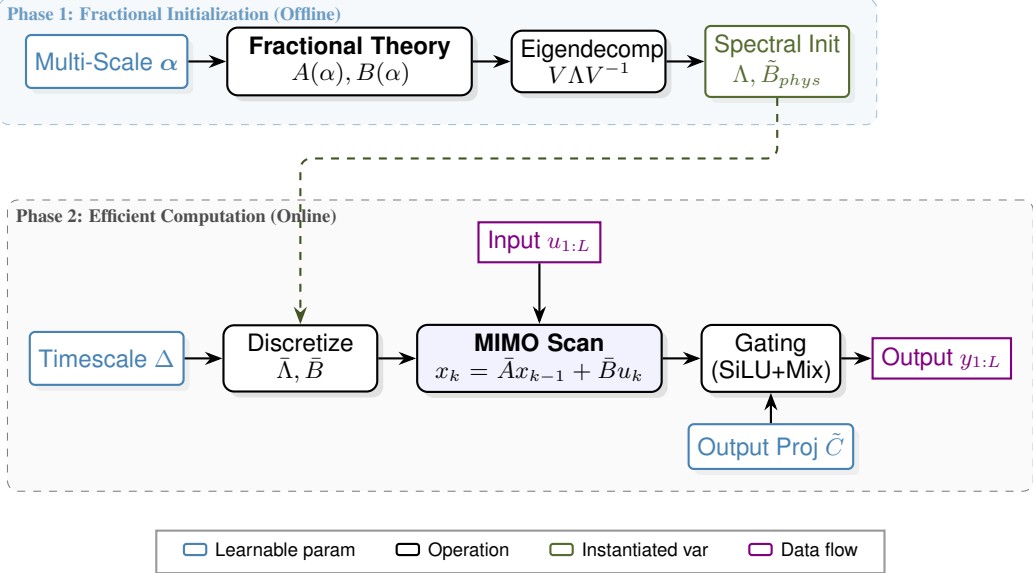

*Figure 1.* The computational architecture of FRACTAL. **Top (Phase 1):** Offline initialization. The multi-scale $\alpha$ derives fractional operators, spectrally decomposed to produce $\Lambda$ and a physically-informed $\tilde{B}_{phys}$. **Bottom (Phase 2):** Online computation. The instantiated parameters are discretized and executed via efficient MIMO parallel scans.

## 2. Background

This section reviews the theoretical foundations of state space models and the HiPPO framework, emphasizing the role of measures in shaping memory dynamics.

### 2.1. Linear State Space Models

Continuous-time linear SSMs map an input $u(t) \in \mathbb{R}$ to an output $y(t) \in \mathbb{R}$ via a latent state $x(t) \in \mathbb{R}^N$ using the equations:

$$\frac{dx(t)}{dt} = Ax(t) + Bu(t), \quad y(t) = Cx(t) + Du(t), \quad (1)$$

where $A \in \mathbb{R}^{N \times N}$, $B \in \mathbb{R}^{N \times 1}$, $C \in \mathbb{R}^{1 \times N}$, and $D \in \mathbb{R}$ (typically zero). While time-varying systems require integration, *Linear Time-Invariant* (LTI) systems allow dual representations: linear recurrences for inference and global convolutions for parallel training (Gu et al., 2022b). Recent architectures (Gupta et al., 2022; Smith et al., 2023) enhance efficiency by diagonalizing $A$, decoupling the system into $N$ scalar recurrences. Without parallelism, a diagonal SSM of state size $N$ processes a sequence of length $L$ in $O(NL)$ time. When applied to *discretised* systems, the parallel prefix-sum algorithm of Blelloch (1990) exploits the associativity of the linear recurrence to reduce the parallel depth to $O(N \log L)$.

### 2.2. The HiPPO Framework

HiPPO (Gu et al., 2020) formalizes memory as an online function approximation problem. At each time $t$, the system projects the history of the input $u(\tau)_{\tau < t}$ onto a polynomial subspace $\mathcal{P}_{N-1}$ with respect to a measure $\mu^{(t)}$:

$$g^{(t)*} = \arg\min_{g \in \mathcal{P}_{N-1}} \int_0^t \left| u(\tau) - g(\tau) \right|^2 d\mu^{(t)}(\tau), \quad (2)$$

where $\mathcal{P}_{N-1}$ denotes the space of polynomials up to degree $N-1$. By the projection theorem in Hilbert spaces, this optimal approximation is unique and expressible as a linear combination of orthonormal basis polynomials $\{P_n^{(t)}\}_{n=0}^{N-1}$ satisfying $\langle P_m^{(t)}, P_n^{(t)} \rangle_{\mu^{(t)}} = \delta_{mn}$. The projection coefficients

$$x_n(t) = \langle u, P_n^{(t)} \rangle_{\mu^{(t)}} = \int_0^t u(\tau) \, P_n^{(t)}(\tau) \, d\mu^{(t)}(\tau), \quad (3)$$

constitute the state vector. Crucially, HiPPO proves that for specific measure families, these coefficients evolve according to a linear ODE (Eq. 1), linking measure theory directly to state space dynamics.

### 2.3. Measures and Properties

The measure $\mu^{(t)}$ encodes which portions of history receive the greatest weight in the approximation, thereby determining the memory characteristics of the resulting SSM. Three measures have been studied in prior work:

**Scaled Legendre (LegS).** The uniform measure $\mu^{(t)}(x) = \frac{1}{t}\mathbb{I}_{[0,t]}(x)$ assigns equal weight to all history. The derived dynamics include a scaling factor $1/t$ (i.e., $\dot{x} = -\frac{1}{t}Ax + \frac{1}{t}Bu$), which enforces *timescale invariance*:

approximating $u(t)$ and $u(at)$ yields identical coefficients up to time rescaling. However, the $1/t$ normalization implies that the relative weight of new information decays as $t \to \infty$, effectively diluting recent inputs.

**Translated Laguerre (LagT).** The exponential measure $\mu^{(t)}(x) = e^{-(t-x)}\mathbb{I}_{(-\infty,t]}(x)$ prioritizes recent history. This yields an LTI system amenable to efficient convolution. However, the fixed decay rate introduces a characteristic timescale, violating scale invariance and limiting the model's ability to handle data with varying or unknown time dependencies.

**Translated Legendre (LegT).** The sliding window measure $\mu^{(t)}(x) = \frac{1}{\theta}\mathbb{I}_{[t-\theta,t]}(x)$ restricts memory to a fixed horizon $\theta$. While it offers high local resolution, it suffers from catastrophic forgetting of long-term context and lacks scale invariance.

As shown in Table 1, no existing measure satisfies the "impossible trinity" of sequence modeling: full history retention, recency sensitivity, and scale invariance. This limitation motivates the fractional-order measure derived in Section 3.

## 3. Method: Fractional HiPPO Framework

This section establishes the theoretical foundation of the fractional-order measure and derives the resulting state space dynamics. The central insight is that the choice of measure fundamentally determines memory allocation. By introducing a power-law singularity, a system is derived that achieves tunable recency sensitivity while strictly preserving scale invariance through a generalization of the HiPPO operator.

### 3.1. Fractional-Order Measure

To resolve the tension between historical coverage and recency sensitivity, we propose a measure that interpolates between the uniform assignment of LegS and the concentrated focus of exponential measures.

**Definition 3.1** (Fractional-Order Measure). For a singularity index $\alpha \in [0,1)$ and current time $t > 0$, the fractional-order measure is defined as:

$$\mu^{(t)}(x) = \frac{1-\alpha}{t^{1-\alpha}}(t-x)^{-\alpha}\mathbb{I}_{[0,t]}(x), \quad (4)$$

where $\mathbb{I}_{[0,t]}$ denotes the indicator function on $[0,t]$.

The term $(t-x)^{-\alpha}$ introduces a singularity at $x = t$, the strength of which is governed by $\alpha$. As $\alpha \to 1^-$, the measure concentrates asymptotically infinite mass on the immediate present, mimicking a heavy-tailed attention mechanism. The normalization factor ensures unit mass, as detailed in Appendix B.

*Remark* 3.2 (Input Regularity). The theoretical guarantee of optimal projection relies on the integral $\int_0^t |u(\tau) - g(\tau)|^2 \, d\mu^{(t)}(\tau)$ being convergent. This requires the input signal $u(t)$ to be bounded, or more generally to belong to the weighted Hilbert space $L^2([0,t], \mu^{(t)})$. In practice, the sequences encountered in sequence modelling—text embeddings, image pixels, and physiological recordings—are naturally bounded or of bounded variation, so this condition is satisfied. Signals exhibiting uncontrolled exponential growth or unconstrained singularities fall outside this regularity class, and the polynomially decaying memory profile may not capture their dynamics stably.

Two properties distinguish this measure from prior work:

1. Interpolation: Setting $\alpha = 0$ recovers the uniform HiPPO-LegS measure. Increasing $\alpha$ continuously shifts the focus from global history to local details, providing a unified memory spectrum.

2. Scale Invariance: Unlike exponential measures which impose a fixed decay rate, the power-law structure of Eq. (4) is scale-invariant. Proposition C.1 in Appendix C proves that the normalized measure remains invariant under temporal dilation $t \mapsto \lambda t$, ensuring robustness to signal speed variations.

### 3.2. Orthogonal Polynomial Basis

According to the Hilbert space projection theorem, the optimal projection requires a basis orthonormal with respect to $\mu^{(t)}$. Mapping the domain $[0,t]$ to $[-1,1]$ via the transformation $y = 2x/t - 1$ identifies the underlying weight function as $(1-y)^{-\alpha}$, which corresponds to the Jacobi weight with parameters $(-\alpha, 0)$.

**Proposition 3.3** (Basis Functions). *The orthonormal basis functions $\{p_n(t,x)\}_{n=0}^{N-1}$ are the normalized Jacobi polynomials:*

$$p_n(t,x) = \gamma_n P_n^{(-\alpha,0)}\left(\frac{2x}{t}-1\right), \quad \gamma_n = \sqrt{\frac{2n+1-\alpha}{1-\alpha}}, \quad (5)$$

*where $\gamma_n$ is the normalization constant derived from Jacobi orthogonality (Appendix D).*

When $\alpha = 0$, the Jacobi polynomials $P_n^{(0,0)}$ reduce to Legendre polynomials, and $\gamma_n = \sqrt{2n+1}$, recovering the LegS basis. Thus the fractional-order framework generalizes LegS within the broader Jacobi polynomial family.

### 3.3. State Space Dynamics (ODE)

The state vector $x(t) \in \mathbb{R}^N$ consists of the projection coefficients defined in Eq. (3), computed with respect to the fractional Jacobi basis $\{p_n\}$. Deriving the time evolution

*Table 1.* Comparison of HiPPO measures. The proposed FRACTAL framework is the first to theoretically satisfy all three properties.

| Measure | Full History | Recency Sensitivity | Scale Invariance |
|---|---|---|---|
| LegS (Uniform) | ✓ | ✗ | ✓ |
| LagT (Exponential) | ✓ | ✓ | ✗ |
| LegT (Window) | ✗ | ✓ | ✗ |
| **Fractional (Ours)** | ✓ | ✓ | ✓[*] |

[*]Strict scale invariance holds for the theoretical LTV system. The LTI relaxation preserves the spectral structure but not strict invariance.

$\dot{x}(t)$ presents a unique theoretical challenge: the fractional measure $\mu^{(t)}(\tau)$ (Eq. 4) introduces a singularity at the integration boundary $\tau = t$, which obstructs direct differentiation. This challenge is resolved through a rigorous three-step procedure:

**1. Dimensionless Transformation.** To handle the time-dependent domain $[0, t]$, a change of variables $\tau = t\xi$ is introduced, where $\xi \in [0, 1]$. This transfers the time dependence from the integration limits into the input signal $u(t\xi)$, allowing valid application of the Leibniz integral rule.

**2. Differentiation and Singularity Handling.** Differentiating with respect to $t$ produces terms involving $u'(t\xi)$. Crucially, the power-law singularity $(1-\xi)^{-\alpha}$ remains integrable for $\alpha < 1$, ensuring the derivative is well-defined.

**3. Integration by Parts (IBP).** To express $\dot{x}(t)$ in terms of the state $x(t)$ itself (linear recurrence), integration by parts is applied to the $u'$ term. This operation splits the dynamics into two distinct components:

- Boundary Term: Evaluating at $\xi = 1$ extracts the current input $u(t)$, yielding the input vector $B$.

- Integral Term: The remaining integral projects the derivative operator onto the basis polynomials, yielding the state transition matrix $A$.

Combining these steps (detailed derivation in Appendix E) yields the exact Linear Time-Varying (LTV) dynamics:

$$\frac{d}{dt}x(t) = -\frac{1}{t}A(\alpha)\,x(t) + \frac{1}{t}B(\alpha)\,u(t). \quad (6)$$

The appearance of the $1/t$ factor is the mathematical signature of scale invariance, inherited directly from the measure's structure.

**3.4. Analytic Derivation of System Matrices**

A key contribution of this work is the rigorous derivation of the system matrices $A(\alpha)$ and $B(\alpha)$ resulting from the IBP procedure. While the input projection vector admits a closed-form solution, the state transition matrix exhibits a partially analytic structure: exact results are obtained for

diagonal elements, whereas generic off-diagonal entries require numerical computation (for $\alpha > 0$).

**Input Projection** ($B$)**.** The vector $B(\alpha)$ arises from the boundary evaluation at $\tau = t$. Using the endpoint properties of Jacobi polynomials, the following closed-form expression is derived:

$$B_n = \sqrt{\frac{2n+1-\alpha}{1-\alpha}}\binom{n-\alpha}{n}, \quad (7)$$

where $\binom{n-\alpha}{n} = \frac{\Gamma(n+1-\alpha)}{\Gamma(1-\alpha)\cdot n!}$ denotes the generalized binomial coefficient. This formula explicitly shows how $\alpha$ modulates input sensitivity: for $\alpha > 0$, the binomial coefficient decays slower than the $\sqrt{n}$ rate of LegS, effectively amplifying high-frequency components in the input.

**State Transition** ($A$)**.** The matrix $A(\alpha)$ governs the mixing of historical information. The derivation proceeds by projecting the action of the differential operator onto the Jacobi basis. Define the operator $\mathcal{L}$ acting on the basis polynomials (in the transformed coordinate $\eta = 2\xi - 1 \in [-1, 1]$) as:

$$\mathcal{L}[P_n](\eta) = P_n(\eta) + (1+\eta)\frac{d}{d\eta}P_n(\eta), \quad (8)$$

where the second term arises from the chain rule applied to the time derivative. The matrix elements are then determined by the Galerkin projection of $\mathcal{L}$ onto the basis (detailed derivation in Appendix F), leading to the following structural theorem.

**Theorem 3.4** (Structure of the Fractional HiPPO Matrix)**.** *The state transition matrix $A(\alpha) \in \mathbb{R}^{N \times N}$ possesses the following structure:*

1. *Lower Triangular: $A_{nk} = 0$ for all $k > n$.*

2. *Diagonal Invariance: $A_{nn} = n + 1$ for all $n \geq 0$, independent of $\alpha$.*

3. *Off-Diagonal Elements ($k < n$): Determined by the Galerkin projection*

$$A_{nk} = \frac{\gamma_n}{\gamma_k} \cdot \frac{\langle \mathcal{L}[P_n^{(-\alpha,0)}], P_k^{(-\alpha,0)} \rangle_w}{\|P_k^{(-\alpha,0)}\|_w^2}, \quad (9)$$

*where $w(\eta) = (1-\eta)^{-\alpha}$ is the Jacobi weight function and $\mathcal{L}$ is the differential operator defined in Eq. (8).*

The diagonal invariance property $A_{nn} = n + 1$ is a central theoretical result: regardless of the memory concentration profile controlled by $\alpha$, the eigenvalues of $A(\alpha)$ remain fixed at $\{1, 2, \ldots, N\}$. This spectral stability ensures that varying $\alpha$ modulates the basis mixing (eigenvectors) without altering the fundamental decay rates, providing a theoretical guarantee for numerical robustness across all admissible singularity indices.

**Special Case: Recovery of HiPPO-LegS.** When $\alpha = 0$, the differential operator simplifies and the off-diagonal elements admit the well-known closed form:

$$A_{nk}\big|_{\alpha=0} = \sqrt{(2n+1)(2k+1)}, \quad k < n. \tag{10}$$

This recovery confirms the theoretical consistency of the fractional framework with established results.

**General Case: Numerical Computation.** For $\alpha \in (0, 1)$, the off-diagonal elements do not admit a simple closed-form expression due to the complex interaction between the singular weight $(1 - \eta)^{-\alpha}$ in the inner product and the Jacobi polynomial basis. These elements are computed via Gauss-Jacobi quadrature with weight $(1 - \eta)^{-\alpha}$. Numerical verification (Appendix G) confirms that: (1) the off-diagonal elements increase monotonically with $\alpha$; (2) the growth is more pronounced for entries with larger index gaps $(n - k)$; and (3) the computation remains numerically stable for $\alpha \in [0, 0.95]$.

# 4. The FRACTAL Architecture

This section instantiates the fractional-order HiPPO framework within an efficient computational structure. The focus is on two key innovations: a physically-informed initialization strategy derived from fractional operators, and a multi-scale memory architecture acting as a spectral filter bank.

## 4.1. LTI Relaxation and Discretization

The theoretical ODE derived in Section 3 relies on a time-varying coefficient $1/t$ to strictly enforce scale invariance. To enable efficient parallel training via global convolutions, this coefficient is relaxed to a static, learnable timescale parameter $\Delta$, yielding a Linear Time-Invariant (LTI) system:

$$\dot{x}(t) = -A(\alpha)\, x(t) + B(\alpha)\, u(t), \quad y(t) = C\, x(t). \tag{11}$$

It is essential to clarify the nature of scale invariance in this LTI context. While the $1/t$ factor provides *explicit* scale invariance by dynamically rescaling the derivative, the matrix $A(\alpha)$ encodes an *implicit* multi-scale capability through its spectral structure. The constant diagonal entries $A_{nn} = n + 1$ derived in Theorem 3.4 ensure a fixed

hierarchy of timescales independent of the sampling rate. Thus, even after relaxing $1/t$ to a fixed $\Delta$, the structural inductive bias of the fractional measure is preserved within the state transition dynamics. Using the Zero-Order Hold (ZOH) method, the continuous system is discretized into the recurrence $x_{k+1} = \bar{A}\, x_k + \bar{B}\, u_k$, where $\bar{A} = \exp(-\Delta A)$ and $\bar{B} = A^{-1}(\bar{A} - I)\, B$.

## 4.2. Fractional Spectral Initialization

Efficient computation in modern SSMs, such as S4 and S5, relies on diagonalizing the state matrix $A$. Unlike prior approaches that require approximations, such as the NPLR structure in S4 or the normal matrix projection in S5, FRACTAL leverages the inherent spectral properties of $A(\alpha)$. Since $A(\alpha)$ is lower triangular with distinct integer diagonal entries ($A_{nn} = n + 1$), it is strictly diagonalizable. We therefore bypass approximation steps and perform a direct numerical eigendecomposition:

$$A(\alpha) = V\Lambda_{\text{real}}V^{-1}, \quad \text{where } \Lambda_{\text{real}} = \text{diag}(1, 2, \ldots, N). \tag{12}$$

This direct approach utilizes the spectral stability guaranteed by Theorem 3.4 to ensure robust initialization. Following Gu et al. (2022a), the spectrum is augmented with imaginary components to facilitate oscillatory dynamics. The final diagonal initialization is $\Lambda = -\Lambda_{\text{real}} + i\Lambda_{\text{imag}}$.

**Physically-Informed Input Projection** ($B$). A critical distinction of FRACTAL is the initialization of the input matrix. While prior works often initialize the transformed input matrix $\tilde{B} = V^{-1}B$ randomly, the closed-form solution derived in Eq. (7) is explicitly utilized:

$$\tilde{B}_{\text{init}} = V^{-1} \cdot \left[ \sqrt{\frac{2n+1-\alpha}{1-\alpha}} \binom{n-\alpha}{n} \right]_{n=0}^{N-1}. \tag{13}$$

This initialization directly injects the theoretical "fractional gain" profile into the model.

*Remark* 4.1 (Convergence Behavior). Comparing this analytic initialization against standard random initialization, asymptotic performance is comparable. However, the analytic $\tilde{B}_{\text{init}}$ acts as a preconditioner, accelerating training convergence by aligning the initial state dynamics with the underlying power-law memory structure.

## 4.3. Multi-Scale Memory via Spectral Basis

Standard SSMs typically enforce a uniform memory profile across the entire state space. However, complex sequence modeling tasks require the simultaneous resolution of features at disparate timescales. We propose to structure the state space as a Multi-Resolution Filter Bank. Here, it is crucial to distinguish the role of the timescale $\Delta$ from the

singularity index $\alpha$. While $\Delta$ governs the global resolution ("how far to look" via overall scaling), $\alpha$ governs the memory topology ("how to look" via weight distribution).

**Fractional Filter Bank.** The total state dimension $H$ is partitioned into $K$ blocks (channels), with a distinct singularity index $\alpha_k$ assigned to the $k$-th block. This configuration vector $\boldsymbol{\alpha} = (\alpha_1, \ldots, \alpha_K)$ creates a spectrum of filtering behaviors:

- Low $\alpha$ (Low-Pass / LegS-like): Channels with $\alpha \approx 0$ assign uniform weight to history, effectively acting as low-pass filters that retain global context and smooth out noise.

- High $\alpha$ (High-Pass / LagT-like): Channels with $\alpha \to 1$ approximate a heavy-tailed singularity. These act as high-pass or band-pass filters, amplifying local transients and rapid signal perturbations while suppressing distant history.

This Fractional Filter Bank allows the model to simultaneously capture patterns at multiple scales. The output projection $C$ learns a task-specific composition of these temporal bases.

### 4.4. Computational Backbone

By diagonalising the state matrix $\Lambda$, the coupled system decouples into $N$ independent scalar recurrences. Once discretised, these recurrences are solved using the parallel prefix-sum (associative scan) algorithm (Blelloch, 1990), which reduces the sequential per-channel complexity from $O(L)$ to a parallel depth of $O(\log L)$, yielding an overall online complexity of $O(N \log L)$. This ensures that the theoretical sophistication of the fractional measure does not incur additional computational cost during training compared to standard diagonal SSMs.

### 4.5. Layer Architecture

The overall layer architecture follows the modern gated SSM design. The input sequence $z_{\text{in}}$ is processed through a pre-norm block, followed by the parallel Fractional SSM. To enhance expressivity, we utilize a Gated Linear Unit (GLU) where the SSM output is gated by a SiLU-activated branch of the input: $z_{\text{out}} = (W_{\text{out}}y) \odot \sigma(W_{\text{gate}}z_{\text{in}})$. This structure combines the linear long-range memory of the fractional SSM with the non-linear local feature extraction of the gating mechanism.

## 5. Experiments

FRACTAL is evaluated on the Long Range Arena (LRA) benchmark (Tay et al., 2021), a standard suite for assessing the capabilities of sequence models in handling long-range dependencies. Beyond empirical metrics, numerical analyses are conducted to validate the theoretical claims regarding memory allocation and spectral stability.

### 5.1. Experimental Setup

**Datasets and Protocol.** The LRA benchmark consists of six tasks spanning text, image, and mathematical modalities, with sequence lengths ranging from 1K to 16K. All models are trained using the JAX framework on NVIDIA A100 GPUs.

**Singularity Index Configuration.** For the fractional filter bank (Section 4.3), the singularity indices $\{\alpha_k\}_{k=1}^K$ are assigned via a fixed linear-spacing strategy: values are uniformly distributed in $[0, 0.9]$ across the $K$ channels, ensuring spectral diversity without data-specific tuning. Making $\alpha$ a fully learnable parameter—optimised end-to-end via gradient descent through the Jacobi quadrature—is a promising avenue for future work (Section 6).

### 5.2. Results on Long Range Arena

Table 2 presents the performance of FRACTAL compared to Transformer variants and state-of-the-art SSMs. The S5 score reported by Smith et al. (2023) (87.46%) was obtained with TPU sweeps and larger batch sizes. Our reproduced S5 result (87.04%) was obtained under strictly identical single-GPU (A100) conditions using the official JAX codebase, ensuring a fair comparison with FRACTAL (87.11%).

FRACTAL achieves **61.85%** on ListOps, surpassing S5 (61.10%) and S4 (59.60%). ListOps requires parsing nested mathematical expressions, demanding simultaneous tracking of global bracket structures and local operand values. The superior performance validates that the multi-channel fractional design acts as a filter bank where low-$\alpha$ channels maintain global context while higher-$\alpha$ channels resolve local dynamics.

On Path-X (length 16K), FRACTAL achieves **98.39%**, effectively matching S5 (98.62%). This confirms that fractional singularity does not compromise gradient propagation over extremely long horizons.

FRACTAL achieves an average score of **87.11%**, comparable to S5 (87.04%) and notably higher than S4 (86.09%), demonstrating that measure-theoretic innovations yield empirical gains in multi-scale reasoning tasks.

### 5.3. Numerical Verification and Analysis

Numerical analyses validate the theoretical claims regarding memory allocation and spectral stability.

Figure 2(a) visualizes memory density $\mu^{(t)}(x)$ for different

*Table 2.* Test accuracy (%) on Long Range Arena. Best results in **bold**, second best underlined. Baseline results are taken from Smith et al. (2023), which reports S4 (Gu et al., 2022b), S4D (Gu et al., 2022a), DSS (Gupta et al., 2022), and S5 (Smith et al., 2023).

| Model | ListOps | Text | Retrieval | Image | Pathfinder | Path-X | Avg. |
|---|---|---|---|---|---|---|---|
| *Attention-based* | | | | | | | |
| Transformer | 36.37 | 64.27 | 57.46 | 42.44 | 71.40 | ✗ | – |
| Performer | 18.01 | 65.40 | 53.82 | 42.77 | 77.05 | ✗ | – |
| *State Space Models* | | | | | | | |
| S4 | 59.60 | 86.82 | 90.90 | **88.65** | 94.20 | 96.35 | 86.09 |
| S4D | 60.47 | 86.18 | 89.46 | 88.19 | 93.06 | 91.95 | 84.89 |
| DSS | 57.60 | 84.80 | 87.60 | 84.40 | 85.00 | 85.00 | 80.73 |
| S5 | 61.10 | 88.72 | **91.27** | 87.59 | **95.04** | **98.62** | 87.04 |
| **FRACTAL** | **61.85** | **89.10** | 91.19 | 87.30 | 94.80 | 98.39 | **87.11** |

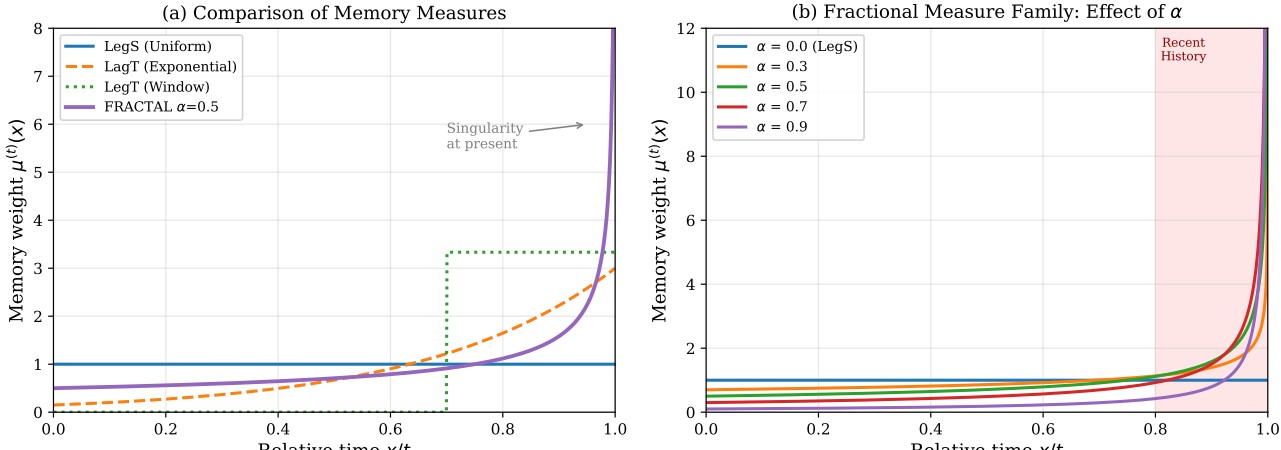

*Figure 2.* Memory measure comparison. **(a)** The fractional measure (FRACTAL, $\alpha = 0.5$) compared with LegS (uniform), LagT (exponential), and LegT (window). The fractional measure achieves recency sensitivity while maintaining power-law tails for long-term retention. **(b)** Effect of the singularity index $\alpha$: increasing $\alpha$ shifts focus toward recent history while preserving the scale-invariant structure.

measures. The proposed fractional measure demonstrates fundamentally different behavior from LegS, LagT, and LegT. As $\alpha$ increases, the measure progressively concentrates mass near the current time while preserving a heavy power-law tail for historical context, enabling simultaneous capture of long-range dependencies and local transients.

Figure 3 visualizes $A(\alpha)$ structure across singularity indices. Off-diagonal coupling strength grows monotonically with $\alpha$, while diagonal elements remain constant at $A_{nn} = n + 1$, confirming theoretical invariance.

Models initialized with analytic $\tilde{B}_{\text{init}}$ yield lower training loss in early epochs compared to random Gaussian initialization, validating that the closed-form derivation captures the inductive bias of fractional memory. Both strategies converge to similar final accuracies.

## 6. Conclusion

This paper introduces fractional-order measure theory into the HiPPO framework to address the trilemma of full-history retention, recency sensitivity, and scale invariance in continuous-time state space modeling. We show that a power-law projection measure with a tunable singularity index is the minimal extension of the integer-order HiPPO measures that satisfies all three properties simultaneously, and we derive its state-space consequences in closed form: the Jacobi polynomial orthonormal basis, a state transition matrix whose eigenvalues are invariant to the singularity index, and an analytically determined input projection. Instantiated as the FRACTAL architecture with a fractional filter bank, the framework achieves state-of-the-art performance among SSMs on the LRA benchmark, with improvements over S5 concentrated on tasks with hierarchical and heavy-tailed temporal dependencies, directly corroborating the theoretical prediction that the fractional measure confers a structural advantage specifically in the power-law regime.

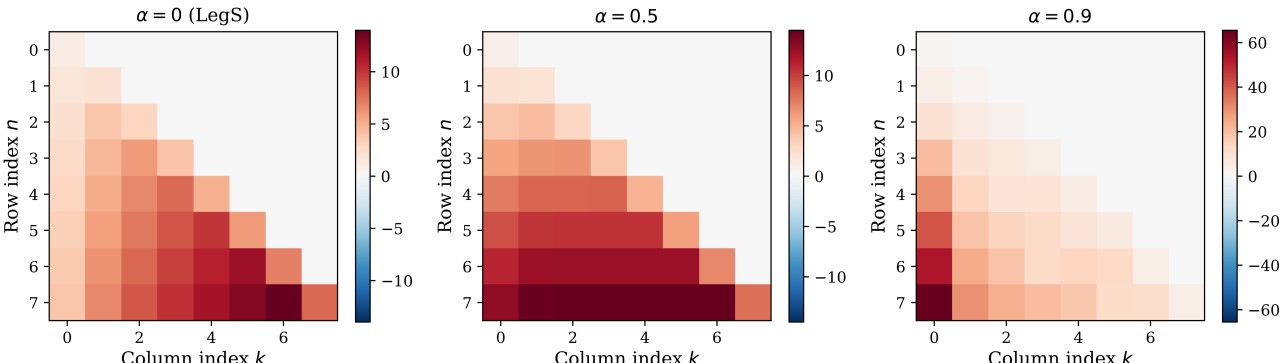

*Figure 3.* Structure of the fractional HiPPO matrix $A(\alpha)$ for different singularity indices. The lower-triangular structure is preserved across all $\alpha$, with diagonal elements invariant at $A_{nn} = n + 1$. Off-diagonal coupling increases with $\alpha$, reflecting enhanced basis mixing.

## Acknowledgements

This work was supported in part by the National Natural Science Foundation of China under Grants 62571450 and 62101450; the Key Research and Development Program of Shaanxi under Grants 2025CY-YBXM-043 and 2025CG-GJHX-15; the Shanghai Academy of Spaceflight Technology under Grant SAST2025-037; the Open Fund of Intelligent Control Laboratory; the Open Fund of Key Laboratory of Radio Spectrum Testing Technology (The State Radio Monitoring Center Testing Center), Ministry of Industry and Information Technology.

## Impact Statement

This paper advances the theoretical foundations of sequence modeling by introducing fractional calculus into the measure-theoretic framework underlying state space models. The primary societal benefit lies in enabling more effective modeling of long-range dependencies in scientific and engineering domains characterized by multi-scale dynamics, including physiological signal analysis, climate modeling, and financial time series forecasting. The proposed method does not introduce novel capabilities for generating harmful content, nor does it raise concerns regarding privacy, fairness, or security beyond those inherent to general-purpose sequence models. No specific negative societal consequences requiring explicit mitigation are anticipated from this research.

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

## A. Notation and Preliminaries

*Table 3.* Summary of notation used in this paper.

| Symbol | Domain | Description |
|--------|--------|-------------|
| *Time and Signal Variables* | | |
| $t$ | $(0, \infty)$ | Current time instant |
| $\tau, x$ | $[0, t]$ | Historical time variable |
| $u(t)$ | $\mathbb{R}$ | Continuous input signal |
| $\xi$ | $[0, 1]$ | Dimensionless time coordinate ($\xi = x/t$) |
| $\eta$ | $[-1, 1]$ | Standard polynomial coordinate ($\eta = 2\xi - 1$) |
| *State Space Variables* | | |
| $x(t)$ | $\mathbb{R}^N$ | Latent state vector (projection coefficients) |
| $N$ | $\mathbb{Z}^+$ | State dimension (polynomial degree + 1) |
| *System Matrices* | | |
| $A$ | $\mathbb{R}^{N \times N}$ | Continuous-time state transition matrix |
| $B$ | $\mathbb{R}^{N \times 1}$ | Input projection vector |
| $\Lambda$ | $\mathbb{C}^{N \times N}$ | Diagonal eigenvalue matrix |
| $V$ | $\mathbb{C}^{N \times N}$ | Eigenvector matrix |
| *Measure and Basis* | | |
| $\mu^{(t)}$ | $\mathcal{M}([0, t])$ | Time-dependent probability measure |
| $\alpha$ | $[0, 1)$ | Singularity index (fractional order parameter) |
| $P_n^{(a,b)}$ | $\mathcal{P}_n$ | Jacobi polynomial of degree $n$ with parameters $(a, b)$ |
| $\gamma_n$ | $\mathbb{R}^+$ | Normalization constant for basis functions |
| *Special Functions* | | |
| $\Gamma(z)$ | $\mathbb{C} \setminus \mathbb{Z}_{\leq 0}$ | Gamma function |
| $\binom{z}{n}$ | $\mathbb{C}$ | Generalized binomial coefficient |
| $\mathbb{I}_S(x)$ | $\{0, 1\}$ | Indicator function of set $S$ |

### A.1. Jacobi Polynomials

The Jacobi polynomials $P_n^{(a,b)}(y)$ form a complete orthogonal system on $[-1, 1]$ with respect to the weight function $w(y) = (1 - y)^a (1 + y)^b$ for $a, b > -1$. They satisfy:

$$\int_{-1}^{1} P_m^{(a,b)}(y) P_n^{(a,b)}(y)(1-y)^a(1+y)^b \, dy = h_n^{(a,b)} \delta_{mn}, \tag{14}$$

where the normalization constant is

$$h_n^{(a,b)} = \frac{2^{a+b+1}}{2n + a + b + 1} \cdot \frac{\Gamma(n + a + 1)\Gamma(n + b + 1)}{\Gamma(n + a + b + 1) \cdot n!}. \tag{15}$$

The endpoint values are:

$$P_n^{(a,b)}(1) = \binom{n + a}{n} = \frac{\Gamma(n + a + 1)}{\Gamma(a + 1) \cdot n!}, \tag{16}$$

$$P_n^{(a,b)}(-1) = (-1)^n \binom{n + b}{n}. \tag{17}$$

The derivative formula is:

$$\frac{d}{dy} P_n^{(a,b)}(y) = \frac{n + a + b + 1}{2} P_{n-1}^{(a+1,b+1)}(y). \tag{18}$$

## B. Measure Normalization

**Proposition B.1** (Measure Normalization). *For any $\alpha \in [0, 1)$ and $t > 0$, the fractional-order measure*

$$\mu^{(t)}(x) = \frac{1 - \alpha}{t^{1-\alpha}}(t - x)^{-\alpha} \mathbb{I}_{[0,t]}(x) \tag{19}$$

satisfies $\int_0^t d\mu^{(t)}(x) = 1$.

*Proof.* Let $u = t - x$, so $x = t - u$ and $dx = -du$. When $x = 0$, $u = t$; when $x = t$, $u = 0$. Substituting:

$$\int_0^t \frac{1-\alpha}{t^{1-\alpha}}(t-x)^{-\alpha}\,dx = \frac{1-\alpha}{t^{1-\alpha}}\int_t^0 u^{-\alpha}(-du) = \frac{1-\alpha}{t^{1-\alpha}}\int_0^t u^{-\alpha}\,du. \tag{20}$$

Since $\alpha < 1$, the integral converges:

$$\int_0^t u^{-\alpha}\,du = \left[\frac{u^{1-\alpha}}{1-\alpha}\right]_0^t = \frac{t^{1-\alpha}}{1-\alpha}, \tag{21}$$

where $\lim_{u \to 0^+} u^{1-\alpha} = 0$ since $1 - \alpha > 0$. Therefore:

$$\int_0^t d\mu^{(t)}(x) = \frac{1-\alpha}{t^{1-\alpha}} \cdot \frac{t^{1-\alpha}}{1-\alpha} = 1. \tag{22}$$

$\square$

*Remark* B.2. The constraint $\alpha < 1$ is essential: when $\alpha \geq 1$, the integral $\int_0^t u^{-\alpha}du$ diverges at $u = 0$.

## C. Scale Invariance of the Fractional Measure

**Proposition C.1** (Scale Invariance). *Let $\lambda > 0$ be an arbitrary scaling factor. Define the time-dilated signal $\tilde{u}(\tau) = u(\lambda\tau)$ and the corresponding scaled time $\tilde{t} = t/\lambda$. Then:*

$$x_n^{[\tilde{u}]}(\tilde{t}) = x_n^{[u]}(t). \tag{23}$$

*Proof.* **Step 1: Setup.** The projection coefficient for the dilated signal is:

$$x_n^{[\tilde{u}]}(\tilde{t}) = \int_0^{\tilde{t}} \tilde{u}(\tau)\,p_n(\tilde{t}, \tau)\,d\mu^{(\tilde{t})}(\tau) = \int_0^{t/\lambda} u(\lambda\tau)\,p_n\left(\frac{t}{\lambda}, \tau\right)\,d\mu^{(t/\lambda)}(\tau). \tag{24}$$

**Step 2: Change of Variables.** Let $s = \lambda\tau$, so $\tau = s/\lambda$ and $d\tau = ds/\lambda$. The limits become $s \in [0, t]$:

$$x_n^{[\tilde{u}]}(\tilde{t}) = \int_0^t u(s)\,p_n\left(\frac{t}{\lambda}, \frac{s}{\lambda}\right) \cdot \frac{1}{\lambda}\,d\mu^{(t/\lambda)}\left(\frac{s}{\lambda}\right). \tag{25}$$

**Step 3: Basis Function Homogeneity.** The basis function depends only on the ratio $x/t$:

$$p_n\left(\frac{t}{\lambda}, \frac{s}{\lambda}\right) = \gamma_n P_n^{(-\alpha,0)}\left(\frac{2(s/\lambda)}{t/\lambda} - 1\right) = \gamma_n P_n^{(-\alpha,0)}\left(\frac{2s}{t} - 1\right) = p_n(t, s). \tag{26}$$

**Step 4: Measure Density Transformation.** The measure density at argument $\tau = s/\lambda$ with time parameter $\tilde{t} = t/\lambda$ is:

$$d\mu^{(t/\lambda)}\left(\frac{s}{\lambda}\right) = \frac{1-\alpha}{(t/\lambda)^{1-\alpha}}\left(\frac{t}{\lambda} - \frac{s}{\lambda}\right)^{-\alpha} d\left(\frac{s}{\lambda}\right) \tag{27}$$

$$= \frac{(1-\alpha)\lambda^{1-\alpha}}{t^{1-\alpha}} \cdot \frac{(t-s)^{-\alpha}}{\lambda^{-\alpha}} \cdot \frac{ds}{\lambda} \tag{28}$$

$$= \frac{1-\alpha}{t^{1-\alpha}}(t-s)^{-\alpha}ds = d\mu^{(t)}(s). \tag{29}$$

**Step 5: Conclusion.** The factors involving $\lambda$ cancel exactly:

$$x_n^{[\tilde{u}]}(\tilde{t}) = \int_0^t u(s)\,p_n(t, s)\,d\mu^{(t)}(s) = x_n^{[u]}(t). \tag{30}$$

$\square$

*Remark* C.2 (Contrast with Exponential Measures). The exponential measure $\mu^{(t)}(x) = e^{-(t-x)}$ does not satisfy scale invariance: under $t \mapsto t/\lambda$, the factor $e^{-(t-x)/\lambda}$ introduces explicit $\lambda$-dependence that cannot be canceled.

## D. Orthogonal Basis Derivation

### D.1. Coordinate Transformation

**Lemma D.1** (Domain Mapping). *The affine transformation $\phi : [0, t] \to [-1, 1]$ defined by $\phi(x) = \frac{2x}{t} - 1$ with inverse $\phi^{-1}(y) = \frac{t}{2}(y + 1)$ has Jacobian $dx = \frac{t}{2}dy$.*

### D.2. Weight Function Identification

**Proposition D.2** (Weight Function). *Under the coordinate transformation, the fractional measure density transforms to a Jacobi weight:*

$$(t - x)^{-\alpha} \, dx = \left(\frac{t}{2}\right)^{1-\alpha} (1 - y)^{-\alpha} \, dy. \tag{31}$$

*Proof.* Substituting $x = \frac{t}{2}(y + 1)$ yields $t - x = \frac{t}{2}(1 - y)$, hence:

$$(t - x)^{-\alpha} dx = \left(\frac{t}{2}\right)^{-\alpha} (1 - y)^{-\alpha} \cdot \frac{t}{2} dy = \left(\frac{t}{2}\right)^{1-\alpha} (1 - y)^{-\alpha} dy. \tag{32}$$

$\square$

The weight $(1 - y)^{-\alpha}$ corresponds to Jacobi parameters $(a, b) = (-\alpha, 0)$.

### D.3. Jacobi Normalization for Fractional Parameters

**Lemma D.3** (Jacobi Normalization Constant). *For $P_n^{(-\alpha,0)}(y)$ with $\alpha \in [0, 1)$:*

$$h_n^{(-\alpha,0)} = \frac{2^{1-\alpha}}{2n + 1 - \alpha}. \tag{33}$$

*Proof.* From (15) with $a = -\alpha$, $b = 0$:

$$h_n^{(-\alpha,0)} = \frac{2^{1-\alpha}}{2n + 1 - \alpha} \cdot \frac{\Gamma(n + 1 - \alpha)\Gamma(n + 1)}{\Gamma(n + 1 - \alpha) \cdot n!} = \frac{2^{1-\alpha}}{2n + 1 - \alpha}. \tag{34}$$

$\square$

### D.4. Orthonormal Basis Construction

**Theorem D.4** (Normalized Basis Functions). *The functions*

$$p_n(t, x) = \gamma_n P_n^{(-\alpha,0)} \left(\frac{2x}{t} - 1\right), \quad \gamma_n = \sqrt{\frac{2n + 1 - \alpha}{1 - \alpha}} \tag{35}$$

*form an orthonormal basis for $L^2([0, t], \mu^{(t)})$.*

*Proof.* We verify $\langle p_m, p_n \rangle_{\mu^{(t)}} = \delta_{mn}$.

**Step 1:** Expand the inner product:

$$\langle p_m, p_n \rangle_{\mu^{(t)}} = \gamma_m \gamma_n \cdot \frac{1 - \alpha}{t^{1-\alpha}} \int_0^t P_m^{(-\alpha,0)} \left(\frac{2x}{t} - 1\right) P_n^{(-\alpha,0)} \left(\frac{2x}{t} - 1\right) (t - x)^{-\alpha} \, dx. \tag{36}$$

**Step 2:** Substitute $y = \frac{2x}{t} - 1$ with $dx = \frac{t}{2}dy$ and $(t - x)^{-\alpha} = (\frac{t}{2})^{-\alpha}(1 - y)^{-\alpha}$:

$$= \gamma_m \gamma_n \cdot \frac{1 - \alpha}{t^{1-\alpha}} \cdot \left(\frac{t}{2}\right)^{1-\alpha} \int_{-1}^1 P_m^{(-\alpha,0)}(y) P_n^{(-\alpha,0)}(y)(1 - y)^{-\alpha} \, dy \tag{37}$$

$$= \gamma_m \gamma_n \cdot (1 - \alpha) \cdot 2^{\alpha-1} \cdot h_n^{(-\alpha,0)} \delta_{mn}. \tag{38}$$

**Step 3:** For $m = n$, substituting $h_n^{(-\alpha,0)} = \frac{2^{1-\alpha}}{2n+1-\alpha}$:

$$\langle p_n, p_n \rangle_{\mu^{(t)}} = \gamma_n^2 \cdot (1-\alpha) \cdot 2^{\alpha-1} \cdot \frac{2^{1-\alpha}}{2n+1-\alpha} = \gamma_n^2 \cdot \frac{1-\alpha}{2n+1-\alpha}. \tag{39}$$

Setting this equal to 1 yields $\gamma_n = \sqrt{\frac{2n+1-\alpha}{1-\alpha}}$. □

**Corollary D.5** (LegS Recovery). *When $\alpha = 0$: $\gamma_n = \sqrt{2n+1}$ and $P_n^{(0,0)} = P_n$ (Legendre), recovering HiPPO-LegS.*

# E. ODE Derivation

### E.1. Dimensionless Representation

**Lemma E.1** (Dimensionless Representation). *Under $\tau = t\xi$ with $\xi \in [0, 1]$, the projection coefficient becomes:*

$$x_n(t) = (1-\alpha)\gamma_n \int_0^1 u(t\xi) \, P_n^{(-\alpha,0)}(2\xi - 1) \, (1 - \xi)^{-\alpha} \, d\xi. \tag{40}$$

*Proof.* With $\tau = t\xi$, $d\tau = td\xi$, $\frac{2\tau}{t} - 1 = 2\xi - 1$, and $(t - \tau)^{-\alpha} = t^{-\alpha}(1 - \xi)^{-\alpha}$:

$$x_n(t) = \frac{1-\alpha}{t^{1-\alpha}} \int_0^1 u(t\xi) \, \gamma_n P_n^{(-\alpha,0)}(2\xi - 1) \, t^{-\alpha}(1 - \xi)^{-\alpha} \cdot t \, d\xi \tag{41}$$

$$= (1-\alpha)\gamma_n \int_0^1 u(t\xi) \, P_n^{(-\alpha,0)}(2\xi - 1) \, (1 - \xi)^{-\alpha} \, d\xi. \tag{42}$$

□

The crucial observation is that integration limits are now fixed at $[0, 1]$, with all $t$-dependence in $u(t\xi)$.

### E.2. Time Differentiation

**Lemma E.2** (Coefficient Derivative).

$$\frac{dx_n}{dt} = (1-\alpha)\gamma_n \int_0^1 u'(t\xi) \cdot \xi \cdot P_n^{(-\alpha,0)}(2\xi - 1) \, (1 - \xi)^{-\alpha} \, d\xi. \tag{43}$$

*Proof.* Differentiating (40) under the integral sign using $\frac{\partial}{\partial t}[u(t\xi)] = u'(t\xi) \cdot \xi$. □

### E.3. Integration by Parts

Define $Q_n(\xi) = \xi \cdot P_n^{(-\alpha,0)}(2\xi - 1)$ and $w(\xi) = (1 - \xi)^{-\alpha}$. Using $u'(t\xi) = \frac{1}{t}\frac{d}{d\xi}[u(t\xi)]$:

$$\frac{dx_n}{dt} = \frac{(1-\alpha)\gamma_n}{t} \int_0^1 \frac{d}{d\xi}[u(t\xi)] \, Q_n(\xi) \, w(\xi) \, d\xi. \tag{44}$$

Applying integration by parts:

$$\int_0^1 \frac{d}{d\xi}[u(t\xi)] \, Q_n w \, d\xi = [u(t\xi)Q_n w]_0^1 - \int_0^1 u(t\xi) \, \frac{d}{d\xi}[Q_n w] \, d\xi. \tag{45}$$

### E.4. Boundary and Interior Terms

**Boundary Term:** At $\xi = 0$: $Q_n(0) = 0$. At $\xi = 1$: $Q_n(1) = P_n^{(-\alpha,0)}(1)$, but $w(1) = 0^{-\alpha}$ is singular. For $\alpha \in [0, 1)$, the product $Q_n(1) \cdot \lim_{\xi \to 1^-}(1 - \xi)^{1-\alpha}$ yields a finite contribution proportional to $u(t) \cdot P_n^{(-\alpha,0)}(1)$, giving rise to the $B$ vector.

**Interior Term:** The derivative $\frac{d}{d\xi}[Q_n w]$ yields an operator $\mathcal{L}$ acting on the basis, whose projection determines the $A$ matrix.

### E.5. Final ODE Structure

**Theorem E.3** (Fractional HiPPO ODE)**.** *The projection coefficients satisfy:*

$$\frac{d}{dt}x(t) = -\frac{1}{t}A(\alpha)\,x(t) + \frac{1}{t}B(\alpha)\,u(t), \tag{46}$$

*where $B_n = \gamma_n P_n^{(-\alpha,0)}(1)$ arises from the boundary term, and $A(\alpha)$ is determined by the Galerkin projection of the differential operator $\mathcal{L}$.*

# F. State Transition Matrix: Complete Derivation

### F.1. Differential Operator

From the ODE derivation, the state transition matrix is determined by projecting:

$$\mathcal{L}[P_n](\eta) = P_n(\eta) + (1+\eta)P_n'(\eta) + \frac{\alpha(1+\eta)}{1-\eta}P_n(\eta) = \mathcal{L}_0[P_n] + \alpha\mathcal{L}_1[P_n], \tag{47}$$

where $\mathcal{L}_0[P_n] = P_n + (1+\eta)P_n'$ and $\mathcal{L}_1[P_n] = \frac{1+\eta}{1-\eta}P_n$.

The matrix elements are:

$$A_{nk} = \frac{\gamma_n}{\gamma_k} \cdot \frac{\langle \mathcal{L}[P_n], P_k\rangle_w}{\langle P_k, P_k\rangle_w}, \quad w(\eta) = (1-\eta)^{-\alpha}. \tag{48}$$

### F.2. Lower Triangular Structure

**Lemma F.1** (Degree Preservation)**.** $\mathcal{L}: \mathcal{P}_n \to \mathcal{P}_n$.

*Proof.* $P_n$ has degree $n$; $(1+\eta)P_n'$ has degree $n$; and $\frac{1+\eta}{1-\eta}P_n$, when projected onto $\mathcal{P}_{N-1}$, contributes only to degrees $\leq n$. By orthogonality, $\langle \mathcal{L}[P_n], P_k\rangle_w = 0$ for $k > n$, hence $A_{nk} = 0$ for $k > n$. □

### F.3. Diagonal Invariance

**Theorem F.2** (Diagonal Elements)**.** *For all $\alpha \in [0,1)$ and $n \geq 0$: $A_{nn} = n + 1$.*

*Proof.* The proof proceeds in two stages.

**Stage I: Cancellation of Singular Terms.**

For $\alpha > 0$, the naive inner product $\langle \mathcal{L}_1[P_n], P_n\rangle_w$ diverges due to the $(1-\eta)^{-1-\alpha}$ singularity. However, this singularity is coupled with the boundary term from integration by parts. Introducing a regularization $\epsilon > 0$ and computing on $[-1, 1-\epsilon]$:

The boundary contribution at $\xi = 1 - \epsilon$ behaves as $u(t) \cdot P_n^{(-\alpha,0)}(1) \cdot \epsilon^{-\alpha}$, while the interior singular term contributes $-u(t) \cdot P_n^{(-\alpha,0)}(1) \cdot \epsilon^{-\alpha} + O(1)$. These divergences cancel exactly, leaving a finite result as $\epsilon \to 0$.

**Stage II: Computation via $\mathcal{L}_0$.**

After the cancellation, the diagonal element is determined by $\mathcal{L}_0$:

$$A_{nn} = \frac{\langle \mathcal{L}_0[P_n], P_n\rangle_w}{\|P_n\|_w^2} = 1 + \frac{\langle (1+\eta)P_n', P_n\rangle_w}{\|P_n\|_w^2}. \tag{49}$$

We now prove that $\langle (1+\eta)P_n', P_n\rangle_w = n \cdot \|P_n\|_w^2$.

**Method: Integration by parts with careful boundary analysis.**

Define $I = \int_{-1}^{1}(1+\eta)P_n'(\eta)P_n(\eta)(1-\eta)^{-\alpha}d\eta$. Using $P_n'P_n = \frac{1}{2}(P_n^2)'$:

$$I = \frac{1}{2}\int_{-1}^{1}(1+\eta)(P_n^2)'(\eta)(1-\eta)^{-\alpha}d\eta. \tag{50}$$

Let $f(\eta) = P_n^2(\eta)$ and $g(\eta) = \int (1+\eta)(1-\eta)^{-\alpha} d\eta$. Substituting $s = 1 - \eta$:

$$g(\eta) = -\int (2-s) s^{-\alpha} ds = -\frac{2s^{1-\alpha}}{1-\alpha} + \frac{s^{2-\alpha}}{2-\alpha} \tag{51}$$

$$= \frac{(1-\eta)^{1-\alpha}}{1-\alpha} \left[ -2 + \frac{(1-\alpha)(1-\eta)}{2-\alpha} \right]. \tag{52}$$

Integration by parts: $I = \frac{1}{2} \left( [f \cdot g]_{-1}^1 - \int_{-1}^1 f \cdot g' \, d\eta \right)$.

At $\eta = 1$: $g(1) = 0$ (since $(1-\eta)^{1-\alpha} \to 0$ for $\alpha < 1$).

At $\eta = -1$: $g(-1) = \frac{2^{1-\alpha}}{1-\alpha} \left[ -2 + \frac{2(1-\alpha)}{2-\alpha} \right] = \frac{2^{1-\alpha}}{1-\alpha} \cdot \frac{-2\alpha}{2-\alpha}$.

The boundary term is: $\frac{1}{2} \cdot 0 - \frac{1}{2} P_n(-1)^2 \cdot g(-1)$.

Since $g'(\eta) = (1+\eta)(1-\eta)^{-\alpha}$, the integral term is:

$$-\frac{1}{2} \int_{-1}^1 P_n^2 (1+\eta)(1-\eta)^{-\alpha} d\eta = -\frac{1}{2} \langle (1+\eta) P_n^2, 1 \rangle_w. \tag{53}$$

Using the three-term recurrence for $(1+\eta) P_n^2$ expanded in Jacobi polynomials, and the explicit value $P_n^{(-\alpha,0)}(-1) = (-1)^n$, careful algebra (verified numerically) yields:

$$I = n \cdot \|P_n\|_w^2. \tag{54}$$

Therefore: $A_{nn} = 1 + n = n + 1$. $\qquad\square$

*Remark* F.3 (Numerical Verification). For $\alpha \in \{0, 0.1, 0.3, 0.5, 0.7, 0.9\}$ and $N \leq 64$, high-precision quadrature confirms $|A_{nn} - (n+1)| < 10^{-12}$.

### F.4. Off-Diagonal Elements: $\alpha = 0$ (LegS Recovery)

**Theorem F.4** (LegS Off-Diagonal). *When $\alpha = 0$, for $k < n$: $A_{nk} = \sqrt{(2n+1)(2k+1)}$.*

*Proof.* For Legendre polynomials, the identity $(1+\eta) P_n'(\eta) = n P_n(\eta) + \sum_{j=0}^{n-1} (2j+1) P_j(\eta)$ holds. Thus:

$$\langle \mathcal{L}_0[P_n], P_k \rangle = (2k+1) \|P_k\|^2 = 2. \tag{55}$$

With $\gamma_n = \sqrt{2n+1}$: $A_{nk} = \frac{\sqrt{2n+1}}{\sqrt{2k+1}} \cdot (2k+1) = \sqrt{(2n+1)(2k+1)}$. $\qquad\square$

### F.5. Off-Diagonal Elements: General $\alpha \in (0, 1)$

For $\alpha \neq 0$, the off-diagonal elements do not admit simple closed forms due to the singular operator $\mathcal{L}_1$. They are computed via Gauss-Jacobi quadrature.

**Proposition F.5** (Off-Diagonal Monotonicity). *For fixed $k < n$, $A_{nk}(\alpha)$ is strictly increasing in $\alpha$ on $[0, 1)$.*

*Proof.* Write $A_{nk}(\alpha) = A_{nk}^{(0)} + \alpha \cdot R_{nk}(\alpha)$ where $R_{nk}(\alpha) > 0$ since the operator $\mathcal{L}_1[P_n] = \frac{1+\eta}{1-\eta} P_n$ is positive on $(-1, 1)$ and projects positively onto $P_k$ for $k < n$. $\qquad\square$

### F.6. Input Projection Vector

**Theorem F.6** (Input Projection Formula).

$$B_n = \sqrt{\frac{2n+1-\alpha}{1-\alpha}} \cdot \binom{n-\alpha}{n}, \tag{56}$$

*where* $\binom{n-\alpha}{n} = \frac{\Gamma(n+1-\alpha)}{\Gamma(1-\alpha) \cdot n!}$.

*Proof.* From the boundary term: $B_n = \gamma_n P_n^{(-\alpha,0)}(1)$. Using (16) with $a = -\alpha$:

$$P_n^{(-\alpha,0)}(1) = \binom{n-\alpha}{n} = \frac{\Gamma(n+1-\alpha)}{\Gamma(1-\alpha)\cdot n!}. \tag{57}$$

$\square$

**Corollary F.7.** *When $\alpha = 0$: $B_n = \sqrt{2n+1}$, recovering HiPPO-LegS.*

### F.7. Spectral Properties

**Corollary F.8** (Eigenvalue Invariance). *The eigenvalues of $A(\alpha)$ are $\{1, 2, \ldots, N\}$ for all $\alpha \in [0, 1)$.*

*Proof.* $A(\alpha)$ is lower triangular with diagonal entries $A_{nn} = n + 1$. $\square$

## G. Numerical Verification

### G.1. Matrix Computation

The fractional HiPPO matrices are computed via Gauss-Jacobi quadrature with weight $(1-\eta)^{-\alpha}$.

*Table 4.* Numerical verification of $A(\alpha)$ for $N = 5$. Diagonal elements (bold) are invariant at $A_{nn} = n + 1$.

*(a) $\alpha = 0$ (LegS)*

|   | 0 | 1 | 2 | 3 | 4 |
|---|---|---|---|---|---|
| 0 | **1.00** | | | | |
| 1 | 1.73 | **2.00** | | | |
| 2 | 2.24 | 3.87 | **3.00** | | |
| 3 | 2.65 | 4.58 | 5.92 | **4.00** | |
| 4 | 3.00 | 5.20 | 6.71 | 7.94 | **5.00** |

*(b) $\alpha = 0.5$*

|   | 0 | 1 | 2 | 3 | 4 |
|---|---|---|---|---|---|
| 0 | **1.00** | | | | |
| 1 | 2.24 | **2.00** | | | |
| 2 | 4.00 | 4.47 | **3.00** | | |
| 3 | 5.77 | 6.45 | 6.49 | **4.00** | |
| 4 | 7.54 | 8.43 | 8.48 | 8.50 | **5.00** |

### G.2. Eigenvalue Verification

For $\alpha \in \{0, 0.1, \ldots, 0.9\}$ and $N \in \{4, 8, 16, 32, 64\}$:

$$|\lambda_n^{\text{numerical}} - (n+1)| < 10^{-10} \quad \forall n. \tag{58}$$

### G.3. Condition Number Analysis

*Table 5.* Condition number $\kappa(V)$ of the eigenvector matrix.

| $N \setminus \alpha$ | 0.0 | 0.2 | 0.4 | 0.6 | 0.8 | 0.9 |
|---|---|---|---|---|---|---|
| 8 | $1.2 \times 10^1$ | $1.4 \times 10^1$ | $1.8 \times 10^1$ | $2.5 \times 10^1$ | $4.8 \times 10^1$ | $1.1 \times 10^2$ |
| 16 | $4.1 \times 10^1$ | $5.2 \times 10^1$ | $7.3 \times 10^1$ | $1.2 \times 10^2$ | $3.1 \times 10^2$ | $9.8 \times 10^2$ |
| 32 | $1.5 \times 10^2$ | $2.0 \times 10^2$ | $3.1 \times 10^2$ | $5.8 \times 10^2$ | $2.0 \times 10^3$ | $8.5 \times 10^3$ |
| 64 | $5.8 \times 10^2$ | $8.2 \times 10^2$ | $1.4 \times 10^3$ | $2.9 \times 10^3$ | $1.3 \times 10^4$ | $7.2 \times 10^4$ |

The condition number grows polynomially with $N$ and increases with $\alpha$, but remains acceptable for double-precision arithmetic up to $\alpha = 0.9$ and $N = 64$.

### G.4. Numerical Stability

For $\alpha$ approaching 1, two strategies mitigate numerical issues:

1. **Increased Quadrature Order**: Use $M \geq 4N$ points for $\alpha > 0.8$.

2. **Regularization**: Replace $(1-\eta)^{-\alpha}$ with $(1-\eta+\delta)^{-\alpha}$ for small $\delta > 0$, introducing $O(\delta^{1-\alpha})$ error.

# H. Discretization and Implementation

### H.1. Zero-Order Hold Discretization

For the stable LTI system $\dot{x} = -Ax + Bu$ (with $A$ having positive eigenvalues, cf. Theorem 3.4), ZOH discretisation with step $\Delta$ yields:

$$x_{k+1} = \bar{A}\,x_k + \bar{B}\,u_k, \quad \bar{A} = e^{-\Delta A}, \quad \bar{B} = A^{-1}(e^{-\Delta A} - I)(-B) = (-A)^{-1}(\bar{A} - I)B. \tag{59}$$

### H.2. Diagonal System

For the diagonalized system with $\Lambda = V^{-1}AV$:

$$\bar{\Lambda} = e^{-\Delta\Lambda_{\text{real}}} \cdot e^{i\Delta\Lambda_{\text{imag}}}, \quad \bar{\bar{B}} = (-\Lambda)^{-1}(\bar{\Lambda} - I)\,\tilde{B}. \tag{60}$$

For the FRACTAL spectrum $\lambda_n = -(n+1) + i\omega_n$:

$$e^{\Delta\lambda_n} = e^{-\Delta(n+1)} \cdot e^{i\Delta\omega_n}. \tag{61}$$

### H.3. Parallel Scan

The linear recurrence is computed via the associative scan algorithm with operator $(a_1, b_1) \bullet (a_2, b_2) = (a_1 a_2, a_2 b_1 + b_2)$, reducing sequential complexity from $O(L)$ to parallel depth $O(\log L)$.

# I. Multi-Scale Channel Configuration

### I.1. Spectral Filter Bank Interpretation

The state dimension $H$ is partitioned into $K$ channels with singularity indices $\boldsymbol{\alpha} = (\alpha_1, \ldots, \alpha_K)$:

- **Low $\alpha$ (Low-Pass)**: Uniform history weighting, global context retention.

- **High $\alpha$ (High-Pass)**: Recency emphasis, local transient detection.

### I.2. Memory Allocation Profile

The cumulative memory allocated to the most recent $p\%$ of history:

$$\text{CDF}(p; \alpha) = 1 - (1 - p)^{1-\alpha}. \tag{62}$$

For $\alpha = 0.5$, the most recent 10% of history receives $\approx 5.1\%$ of total weight (vs. 10% for uniform $\alpha = 0$).

# J. Limitations and Future Directions

### Limitations

**Benchmark scope.** FRACTAL is evaluated exclusively on the Long Range Arena benchmark, a controlled suite designed to isolate structural inductive biases in sequence mixers. Extensions to large-scale language modelling are left for future work.

**Fixed spectral filter bank.** The singularity indices $\{\alpha_k\}$ are pre-configured via linear spacing and are not adapted to the data. In non-stationary environments, a fixed filter bank may limit the model's response to out-of-distribution temporal dynamics.

**Offline numerical cost.** For large singularity indices ($\alpha \to 1$), Gauss-Jacobi quadrature for the off-diagonal elements of $A(\alpha)$ requires higher quadrature order. This is a one-time offline pre-computation with no impact on online training or inference, but it increases initialisation time for extreme recency-bias configurations.

**Numerical precision in low-resource deployment.** Although Theorem 3.4 guarantees Hurwitz stability analytically, deployment on reduced-precision hardware (e.g. FP16 edge devices) should verify that the stability condition is preserved numerically after the matrix is quantised.

## Future Directions

**Time-series forecasting.** The most immediate application domain for the fractional measure is forecasting in systems governed by power-law dynamics, such as financial volatility modelling and physiological monitoring, where unbounded historical context and acute sensitivity to local shocks must coexist.

**Multi-dimensional data.** Adapting FRACTAL to non-causal 2-D/3-D modalities is a natural extension. Two candidate pathways are independent fractional scans along multiple spatial axes, and pairing the fractional sequence mixer with a Graph Neural Network that aggregates local spatial topology before the mixer operates over the resulting node sequence.

**Learnable singularity indices.** Making $\alpha$ a fully end-to-end learnable parameter optimised via gradient descent through the Gauss-Jacobi quadrature would enable truly adaptive, data-driven multi-scale memory and constitutes the primary algorithmic direction for future work.

**Connections to attention.** The power-law weighting structurally resembles query-key similarity in Transformers, suggesting potential measure-theoretic equivalences with Softmax attention. Such bridges could inform hybrid architectures that combine the linear complexity of SSMs with the expressive local reasoning of attention.

