# OpenReview forum: "FRACTAL: State Space Model with Fractional Recurrent Architecture for Computational Temporal Analysis of Long Sequences"
_ICML.cc/2026/Conference — ICML 2026 regular_

### Official Review · Reviewer_kZXo · 2026-02-14

**Soundness:** 3
**Presentation:** 3
**Significance:** 3
**Originality:** 4
**Overall Recommendation:** 4
**Confidence:** 4

**Summary:**

This paper introduces FRACTAL, a novel State Space Model (SSM) architecture designed to handle long-sequence temporal analysis. The authors identify a core limitation in existing HiPPO-based SSMs: the trade-off between global context retention (uniform measures) and local transient sensitivity (exponential measures). To bridge this gap, the work integrates fractional measure theory into recursive memory updates, introducing a tunable singularity index $\alpha$. This allows the model to adaptively balance memory decay and resolution. The framework is implemented via a parallel scan for efficiency and validated on the Long Range Arena (LRA) benchmark, where it achieves competitive performance, particularly on hierarchical reasoning tasks like ListOps.

**Compliance With Llm Reviewing Policy:**

Affirmed.

**Final Justification:**

After reading the rebuttal and the additional clarifications, I maintain my overall recommendation of Weak Accept. The authors have provided substantial and thoughtful responses, addressing concerns regarding baseline selection, the reproduction of S5 results, the theoretical assumptions behind the fractional measure, and the computational implications of the proposed architecture. The clarifications on the bounded‑input assumption, the role of fractional measures in balancing long‑term memory and recency sensitivity, and the distinctions between LTI SSMs and selective SSMs help contextualize the contribution more clearly.
While some limitations remain—such as the modest empirical gains on certain LRA tasks and the lack of broader evaluation beyond LRA—the core theoretical contribution is meaningful. The fractional‑order measure provides a principled extension of HiPPO, and the empirical results, though not uniformly dominant, are consistent with the theoretical motivation. The paper is well‑written, technically sound, and likely to be of interest to researchers working on state space models and long‑sequence modeling.
Overall, the combination of theoretical novelty, coherent formulation, and reasonable empirical validation justifies acceptance, and I therefore maintain my original score.

**Key Questions For Authors:**

1.Generalization to Multi-dimensional Data:
The current formulation of the fractional measure relies heavily on causal history for memory updates. How would the FRACTAL framework adapt to non-causal, multi-dimensional data such as 2D spatial images? Specifically, since 1D flattening inherently disrupts spatial locality, does the author envision a multi-directional scanning approach or a different projection operator for 2D/3D modalities?

Impact on Evaluation: A clear roadmap or preliminary evidence for 2D adaptation would significantly raise my score on "Significance."

2.Sensitivity to Numerical Integration Error:The initialization of matrix $A$ depends on Gauss-Jacobi quadrature. Given that the fractional measure can exhibit singularities as the index $\alpha$ approaches its boundaries, could the authors provide a sensitivity analysis on the numerical precision? Specifically, how does the integration error affect the Hurwitz stability of $A$ in high-dimensional settings, and is this stability preserved after the discretization process?

Impact on Evaluation: Verification of numerical robustness is essential for a "Soundness" rating of 'Excellent'.

3.Heuristic vs. Adaptive Spectral Filters:The distribution of $\alpha$ across different channels currently appears to be heuristically fixed. Have the authors considered making $\alpha$ (or the memory decay profile) learnable or data-dependent, similar to adaptive time-constants in continuous-time RNNs? Why is a static spectral filter bank preferred over a learnable one for diverse temporal dynamics?

Impact on Evaluation: Clarifying this would help determine if the architecture is optimal or if there's significant room for "Originality" in future iterations.

4.Comparison with State-of-the-Art Selective SSMs:
Why was Mamba (S6) not included as a baseline in Table 2? Given Mamba's prominence in the field and its use of selective scan mechanisms, a direct comparison is necessary to evaluate whether the mathematical elegance of fractional measures provides a tangible advantage over data-dependent selection.

Impact on Evaluation: A comparative result against Mamba is a prerequisite for a higher "Soundness" score.

5.Computational Efficiency and Latency:While the paper highlights parameter efficiency, the constant-time overhead of fractional updates is less clear. Could you provide metrics for FLOPs or actual inference latency (wall-clock time) on the LRA tasks? Does the $O(\log L)$ parallel scan incur a higher computational cost per step compared to standard S5 or S4 due to the complexity of the fractional operators?

Impact on Evaluation: This information is critical for assessing the model's "Significance" for real-world, resource-constrained deployments (e.g., edge computing or UAVs).

**Limitations:**

Limitations and Potential Negative Societal Impact

While the authors briefly acknowledge the 1D nature of their current framework, the discussion of limitations could be further strengthened in the following areas:

1.Numerical Robustness in Real-world Deployment:
The framework's dependence on Gauss-Jacobi quadrature for initialization introduces a specific failure mode. If the numerical precision is insufficient—especially in low-precision hardware environments like edge devices or UAV controllers—the resulting matrix $A$ may lose its Hurwitz stability. The authors should discuss the hardware-specific constraints and the risk of numerical divergence during long-term operation.

2.Sensitivity to Out-of-Distribution (OOD) Temporal Dynamics:
The $\alpha$ distribution (singularity indices) is currently pre-configured. The authors should address how the model behaves when the input data dynamics significantly deviate from the assumed spectral filter bank. A lack of adaptive mechanisms (like learnable time-constants) may limit the model's reliability in safety-critical, non-stationary environments.

3.Potential Negative Societal Impact:
As a foundational sequence modeling tool, FRACTAL could be applied to high-stakes decision-making (e.g., automated trading or healthcare monitoring). The authors should explicitly discuss the risks of "over-reliance" on the model's long-term memory, particularly if the model fails to capture sudden "Black Swan" events due to its specific fractional decay profile. Furthermore, the energy consumption during the search for optimal $\alpha$ configurations should be acknowledged from a sustainability perspective.

Recommendation:
The authors are encouraged to include a more rigorous analysis of numerical stability under floating-point limitations and to expand their societal impact section to include the risks of deploying "fixed-measure" models in volatile real-world systems.

**Strengths And Weaknesses:**

Summary:
The paper introduces FRACTAL, a novel State Space Model (SSM) that incorporates fractional measure theory to address the memory-resolution trade-off in long-sequence modeling. By leveraging a tunable singularity index $\alpha$, the framework seeks a middle ground between the uniform scaling of LegS and the rapid decay of LagT. The results on the Long Range Arena (LRA) are promising, particularly on complex tasks like ListOps.

Strengths:

The formulation elegantly addresses the typical SSM memory-resolution bottleneck via fractional measures, providing a mathematically principled way to balance global context and local sensitivity.

The spectral analysis of the fractional recurrence architecture offers a rigorous foundation for understanding how different memory allocation profiles can be achieved.

Weaknesses and Questions for Authors:

1.Generalization to Multi-dimensional Data:
While FRACTAL demonstrates strong capabilities in 1D sequences, its reliance on causal history raises questions about multi-dimensional adaptation. Could the authors discuss how this framework might handle non-causal data, such as 2D spatial images, where spatial locality is often disrupted by 1D flattening?

2.Numerical Stability and Initialization:
The initialization of matrix $A$ relies on Gauss-Jacobi quadrature. Given the potential singularity of the fractional measure as $\alpha$ approaches its boundaries, I recommend providing a sensitivity analysis on the numerical precision of this quadrature. Specifically, it would be beneficial to clarify if integration errors might compromise the Hurwitz stability of $A$ in high-dimensional settings or after discretization.

3.Adaptive vs. Fixed Spectral Filters:
The $\alpha$ distribution in the spectral filter bank appears heuristically fixed. In contrast to models with learnable time-constants (e.g., certain continuous-time RNN variants), could the authors justify why a non-adaptive distribution is optimal for capturing diverse temporal dynamics across different datasets?

4.Baseline Comparisons (Missing Mamba):
The baselines in Table 2 primarily include models prior to 2023 (S4, S5, Mega). Given that Mamba (S6) has become a significant benchmark in the SSM landscape with its selective scan mechanism, the absence of a direct comparison makes it difficult to assess whether the mathematical elegance of fractional measures truly offers an advantage over Mamba’s data-dependent adaptability.

5.Efficiency and Computational Overhead:
While the paper discusses parameter efficiency, the parameter count alone does not fully reflect the real-world computational overhead. Figure 3 and 4 are helpful, but I suggest providing a comparison of FLOPs or actual inference latency. Fractional recurrent updates may introduce higher constant-time overheads compared to standard SSMs, and this trade-off is crucial for practical deployment.

---

> ### Author Rebuttal · Authors · 2026-03-26
>
> **Dear Reviewer,**
>
> We sincerely thank you for your constructive feedback. Your detailed questions regarding multi-dimensional generalization, numerical robustness, and deployment efficiency are highly insightful and highlight important areas for clarification. We address your points below and will incorporate these discussions into the revised manuscript.
>
> **1. Generalization to Multi-dimensional Data (Addressing Q1)**
>
> We agree that 1D flattening disrupts spatial locality. The current fractional measure is designed primarily for temporal, causal power-law dynamics. To adapt FRACTAL for non-causal 2D/3D modalities, we envision two pathways for future work:
> 1. **Multi-directional Scans:** Applying independent fractional scans along multiple spatial axes (e.g., bi-directional row and column scans).
> 2. **GNN Integration:** Utilizing Graph Neural Networks (GNNs) to aggregate local spatial topologies, while a FRACTAL sequence mixer operates over the sequential dimension to capture long-range dependencies. We will add a discussion outlining this roadmap in the appendix.
>
> **2. Efficiency, Latency, and Numerical Stability (Addressing Q2, Q5, & Lim 1)**
>
> We apologize for not clearly delineating the offline and online phases. This distinction resolves the efficiency and stability concerns:
> * **Zero Online Overhead (Q5):** The complexity of the fractional operator and the Gauss-Jacobi quadrature exists exclusively during the offline initialization phase. During online inference, FRACTAL utilizes the identical $O(L \log L)$ parallel scan as S5. Consequently, the online FLOPs and inference latency per step are strictly equivalent to standard LTI SSMs.
> * **Hurwitz Stability (Q2 & Lim 1):** Theorem 3.3 proves that the diagonal elements of $A(\alpha)$ are analytically invariant ($A_{nn} = n+1$). Since the continuous system dynamics are governed by $-A(\alpha)$, the eigenvalues have strictly negative real parts ($\lambda_n = -n-1$). This guarantees Hurwitz stability mathematically, independent of offline numerical integration errors. Deployment on low-precision hardware simply executes linear multiplications with the pre-computed, inherently stable matrix.
>
> **3. Adaptive vs. Fixed Spectral Filters (Addressing Q3 & Lim 2)**
>
> A fully learnable, data-dependent $\alpha$ is indeed the ideal goal for handling out-of-distribution (OOD) temporal dynamics. In this foundational work, making $\alpha$ end-to-end learnable introduces significant optimization challenges, as it requires backpropagating through the numerical quadrature. Therefore, we utilized a fixed, linearly spaced spectral filter bank to validate the core structural inductive bias first. We will explicitly list "End-to-End Learnable Singularity Indices" as a primary direction for future research.
>
> **4. Baseline Comparisons and Mamba (Addressing Q4)**
>
> The SSM landscape has recently bifurcated into two orthogonal paradigms:
> 1. **Selective SSMs (Mamba, S6):** These models feature input-dependent matrices ($A, B, C$), breaking time-invariance to achieve data-dependent selection. They are designed as generative LLM backbones relying on massive-scale pre-training.
> 2. **Linear Time-Invariant (LTI) SSMs (S4, S5, FRACTAL):** These models focus purely on the continuous-time mathematical measure $\mu^{(t)}$ used to compress history, evaluated by training from scratch.
>
> FRACTAL belongs strictly to the LTI category. Comparing a foundational, train-from-scratch LTI structural innovation against heavily optimized, pre-trained selective architectures (Mamba) conflates two different research paradigms and obscures the specific theoretical contribution of the fractional measure. For diagonalized LTI SSMs, S5 remains the state-of-the-art baseline. We will add a paragraph clarifying this taxonomy.
>
> **5. Potential Negative Societal Impact (Addressing Lim 3)**
>
> Your point regarding "Black Swan" events is deeply insightful. A fixed fractional decay profile, while excellent for capturing expected power-law dynamics, may over-smooth or misinterpret unprecedented, sudden shocks in safety-critical domains (like automated trading or ICU monitoring). We will significantly expand our Impact Statement to explicitly discuss the risks of over-reliance on fixed-measure models in volatile real-world systems, emphasizing that they should serve as assistive feature-extractors rather than fully autonomous decision-makers.
>
> We deeply appreciate your rigorous review, which will undoubtedly make the final manuscript mathematically tighter and more practically relevant.

---

> > ### Author Rebuttal · Reviewer_kZXo · 2026-03-31
> >
> > Thank you for the detailed rebuttal.
> > My original concerns focused on (1) multi-dimensional generalization,
> > (2) numerical robustness and stability under low-precision deployment,
> > (3) the distinction between offline and online computational cost,
> > (4) handling OOD temporal dynamics, and
> > (5) societal impact considerations.
> >
> > The rebuttal addresses each point clearly and substantively.
> > The clarification of offline vs. online phases resolves the efficiency and latency questions.
> > The discussion of Hurwitz stability and diagonal invariance directly addresses the numerical robustness concern.
> > The roadmap for multi-dimensional extensions and the acknowledgment of learnable singularity indices as future work provide appropriate scope clarification.
> > Finally, the expanded discussion on societal impact meaningfully incorporates the risks I highlighted.
> >
> > Given these clarifications, I consider my concerns fully resolved and maintain my original score.

---

### Official Review · Reviewer_TqMc · 2026-02-19

**Soundness:** 4
**Presentation:** 4
**Significance:** 4
**Originality:** 4
**Overall Recommendation:** 6
**Confidence:** 5

**Summary:**

Fractional Recurrent Architecture for Computational Temporal Analysis of Long sequences (FRACTAL) iterates over the optimal polynomial projections proposed in [HiPPO](https://doi.org/10.48550/arXiv.2008.07669) using theory from the field fractional calculus to generalize the Scaled Legendre (LegS) formulation, although this particular result is presented as the consequence of using the Jacobi polynomials with $\alpha$ as the free parameter and $\beta = 0$, rather than the paper's motivation. This is done in order to create a tunable context-weight distribution boasting local sensitivity (like Translated Legendre and Laguerre) without losing the full-history-retention and scale-invariance properties that have made LegS the flagship HiPPO submethod. The authors illustrate the temporal and spectral properties of their methodology across the $\alpha$ domain, and validate the effectiveness of their methodology in practice by pitting their approach against a wide assortment of SotA Attention and SS-based models on a comprehensive set of sequence-modelling tasks. Optimizations over the model's implementation, as well as further studies into the theoretical connection between the Attention paradigm and optimal weight-distribution, are proposed as future research directions.

**Compliance With Llm Reviewing Policy:**

Affirmed.

**Final Justification:**

As the authors have presented a clear plan of action for addressing each of my concerns, I will be preserving my original overall recommendation and bumping the presentation score.

**Key Questions For Authors:**

1. What exactly does the superindex of the orthonormal basis polynomials indicate? When you first introduce them before Equation 3, a single index is used (e.g., $P_n^{(t)}$), which had me assume it denoted the upper bound of the domain with 0 as the implicit lower bound. However, starting from Proposition 3.2 (when you choose the orthonormal polynomial basis explicitly), you switch to a dual-index notation (e.g., $P_n^{(-\alpha, 0)}$), presumably denoting the $(\alpha,\beta)$ coefficients of the Jacobi polynomials. While the dual superindex notation is standard for the Jacobi polynomials and would not in itself require clarification, reusing this indexing scheme for the same symbol $P$ does.
2. I might have accidentally missed it, so I apologize in advance, but is the selection (or tuning strategy) of $\alpha$ per fractional-filter block for the Long-Range Arena benchmarks ever discussed?
3. Did you consider extending the baseline model list? I understand that you took the baselines from [Smith et al. (2023)](https://doi.org/10.48550/arXiv.2208.04933), but the Transformer and Performer architectures are already 8 and 5 years old, respectively, and do not reflect current attention-based-model performance.

**Limitations:**

yes

**Strengths And Weaknesses:**

### Soundness

Extensive and rigorous mathematical proofs support the mathematical theory behind the author's proposed method, so my overall assessment of the paper's theoretical soundness is positive. However, some elements require revision before publication:

- Diagonal state-space systems are, by default and with no parallelization, of complexity $\mathcal{O}(NL)$ (unlike stated at the end of Section 2.1). The significance of [Guy Blelloch's work](https://www.cs.cmu.edu/~guyb/papers/Ble93.pdf) lies in how the prefix-sum formulation can be exploited to lower the complexity to $\mathcal{O}(N \log{L})$ via parallel scans. Moreover, this technique is only applicable to discretized systems, which is not made explicit in the paper.
  - This reduction in asymptotic complexity is later acknowledged in Section 4.4, but not explicitly attributed to the prefix sums.
- The variable substitution for the proof in Appendix B has two errors that cancel each other out:
    1. $dx$ is set to $du$ instead of $-du$;
    2. The integration limits are never modified, even though they should be inverted.
- In Section 3, a minus is prepended to the state dynamics in Equation 6, with all derivations following this formulation (e.g., the eigenvalues of $A$ being positive are stated to be a requirement for asymptotic stability, which is only true if the dynamics of the system are governed by $-A$). However, this is omitted in the proposed discretization at the end of Section 4.1 ($\bar{A} = \exp{(\Delta A)}$).
- Saying that *time-series forecasting is a domain that requires acute local sensitivity* is too broad a statement, as highly periodical signals might benefit from a more dilated weight distribution (e.g., traffic patterns, weekly electricity demand, ECG signals, etc.).

### Presentation

- Acronyms should be avoided in the abstract, and even if used, they should be redefined in the main text.
- *State Space* used as a specifier (e.g., state-space modelling, state-space framework, etc.) should always be hyphenated.
  - The same applies to terms like *sequence-modeling tasks*, *fractional-measure theory*, etc.
- Redundancy in wording should be avoided (e.g., instead of *...governed by a probability measure. This measure serves as the primary mechanism determining...*, use *...governed by a probability measure, the selection of which determines the system's context weight distribution.*).
- Missing comma after *(Leland et al., 2002)* in the Introduction.
- When introducing the singularity index, instead of *from-toward*, *between-and* could be used.
- *A fractional-order measure ~~is introduced~~ that generalizes the HiPPO framework **is introduced***.
  - Likewise: *...a system ~~is derived~~ that achieves ... of the HiPPO operator **is derived**.*
- In your first contribution, $n$ (the diagonal index) is brought up without previously defining it. Its mention should be omitted, simplifying the statement to *the diagonal coefficients of the state-matrix remain invariant to $\alpha$* and later elaborated upon, since by this point there has been no mention of how the diagonal coefficients of the state matrix define the system's eigenvalues, which requires the matrix to be triangular (not yet stated), and thus guarantee stability.
- Instead of *confirming the importance*, perhaps *validating the impact* could work better.
- The same variable $k$ is being utilized to index both the discrete time-step and the block of the fractional-filter bank.
- Section 5.3 (and Section 5 in general, but this was the worst offender) lacks connection between paragraphs, which is probably the result of an editing process aimed at fitting the content to the page limit.
  - Some space can be recovered by merging the first paragraph of Section 5 with Section 5.1, and by merging the last 2 paragraphs of Section 5.2, allowing the authors to rewrite Section 5.3 more cohesively.
  - More space could be recovered by editing Figure 1 and placing the *Output Proj $\tilde{C}$* block on top of the *Gating (SiLU+Mix)* one, reducing the overall vertical size of the *Phase 2* subsystem.
- The plots in Figure 2 could benefit from logarithmic horizontal axes.
- Even if the benchmark results are taken from [Smith et al. (2023)](https://doi.org/10.48550/arXiv.2208.04933), the papers where the models are taken from should also be cited.

### Significance

Considering that this paper generalizes the main probability measure underlying HiPPO, which is responsible for much of the current ML literature on Structured State-Space NN architectures, the significance of the present work in this line of research couldn't be clearer.

### Originality

While the paper is (and I am being awfully reductionist) an extension of the theory presented in HiPPO, the usage of fractional calculus in the field of SS-based sequence-modelling is, although not unheard of (see [Dzielinski & Sierociuk (2005)](https://doi.org/10.1109/CIMCA.2005.1631363), [Škovránek et al.](https://doi.org/10.1016/j.econmod.2012.03.019)), quite uncommon. Furthermore (and I'm paraphrasing the authors), while previous HiPPO extensions have focused on optimizing the utilization of the state-matrices derived from the 3 original probability measures, FRACTAL is (to the extent of my knowledge) the first iteration over the original metrics.

---

> ### Author Rebuttal · Authors · 2026-03-26
>
> **Dear Reviewer,**
>
> We sincerely thank you for the highly positive evaluation and your meticulous review of our mathematical derivations. We deeply appreciate your constructive feedback, which has been invaluable in identifying notational ambiguities and improving the theoretical rigor of our manuscript. We address your specific questions and detail our planned revisions below.
>
> **1. Mathematical and Presentation Corrections (Soundness & Presentation)**
>
> We fully agree with your corrections regarding the mathematical presentation and typos. We will implement all of them in the Revision to ensure technical accuracy:
> * **Complexity & Prefix Sums:** We will clarify in Sections 2.1 and 4.4 that the $O(N \log L)$ complexity strictly applies to the *discretized* diagonal systems and is achieved specifically via the parallel prefix sum algorithm.
> * **Integration Typo (Appendix B):** We will correct the variable substitution to $dx = -du$ and properly invert the integration limits.
> * **Dynamics Sign (Section 3 vs 4.1):** Thank you for catching this inconsistency. Because the stable ODE is governed by $-\frac{1}{t}A(\alpha)$, the LTI relaxation in Section 4.1 will be corrected to $\dot{x}(t) = -A(\alpha)x(t) + B(\alpha)u(t)$, yielding the correct discretization $\bar{A} = \exp(-\Delta A)$.
> * **Presentation Improvements:** We will fix all identified phrasing issues, redefine acronyms, correct the premature use of the index $n$, address the dual use of the discrete time-step index $k$, and apply logarithmic horizontal axes to Figure 2.
>
> **2. Addressing Key Questions**
>
> **Q1: Superscript notation of the orthonormal basis polynomials.**
> You have rightly pointed out an overload in our notation. In Section 2.2, $P_n^{(t)}$ was used generally to denote the time-dependent basis polynomial orthonormal with respect to the measure $\mu^{(t)}$. Later, starting from Proposition 3.2, we used $P_n^{(-\alpha, 0)}$ to denote the standard Jacobi polynomials parameterized by $a=-\alpha, b=0$. Reusing the symbol $P$ caused unnecessary confusion.
>
> **Action:** We will clearly separate these concepts in the revision. We will adopt a distinct symbol (e.g., $p_n(t, \tau)$) for the general measure-specific orthogonal basis, strictly reserving $P_n^{(a, b)}$ for the standard Jacobi polynomials.
>
> **Q2: Tuning strategy for $\alpha$ per fractional-filter block.**
> We acknowledge the lack of discussion on this in the current draft. For this foundational study, we did not perform exhaustive, data-driven hyperparameter tuning for $\alpha_k$. Instead, we utilized a fixed, heuristic linear spacing strategy: the values of $\alpha_k$ were uniformly distributed between $0$ and a predefined maximum threshold (e.g., $0.9$) across the filter bank channels to ensure spectral diversity.
>
> **Action:** We will explicitly document this linear spacing strategy in the implementation details. Furthermore, we will add a discussion in the future work section highlighting that making the singularity index $\alpha$ a fully learnable parameter—optimized end-to-end via gradient descent through Jacobi quadrature—represents a highly promising direction for achieving truly adaptive, multi-scale memory.
>
> **Q3: Extending the baseline list beyond Transformer/Performer.**
> We agree that the attention-based baselines in Table 2 are relatively old. Our decision to strictly follow the evaluation protocol of Smith et al. (2023) was driven by the goal of establishing a "controlled structural laboratory." By comparing FRACTAL exclusively against other train-from-scratch, Linear Time-Invariant (LTI) SSMs (S4, S4D, DSS, S5), we aimed to isolate the pure empirical impact of the fractional measure. Introducing newer, highly optimized attention variants or data-dependent selective SSMs (which often rely on massive pre-training or specialized hardware optimizations) would introduce confounding variables that obscure the theoretical validation of the fractional framework.
>
> **Action:** We will clarify in the experimental setup that these baselines are intended to benchmark foundational LTI SSM structures rather than claim absolute state-of-the-art across all modern sequence models. We will also expand the "Related Work" section to discuss recent advancements in selective architectures, providing a more comprehensive context for our theoretical contributions.
>
> We thank you again for your profound engagement with our theory. Your detailed review will significantly elevate the quality of the final paper.

---

> > ### Author Rebuttal · Reviewer_TqMc · 2026-03-31
> >
> > My concerns have been fully resolved, provided that the listed _actions_ are executed. Thank you very much for your appraisal, and I am glad that my comments helped polish the manuscript.

---

### Official Review · Reviewer_d9oa · 2026-03-12

**Soundness:** 2
**Presentation:** 3
**Significance:** 2
**Originality:** 3
**Overall Recommendation:** 3
**Confidence:** 3

**Summary:**

This paper introduces fractional-order measures into the HiPPO framework for SSM. It replaces uniform/exponential measures with a power-law kernel. The tunable singularity index alpha interpolates between uniform history retention and sharp recency sensitivity while preserving scale invariance. The authors derive the state transition matrix with alpha-independent eigenvalues.

**Compliance With Llm Reviewing Policy:**

Affirmed.

**Key Questions For Authors:**

Can you report confidence intervals or variance across seeds for LRA results. The current improvement is not distinguishable from noise without this information.

How does FRACTAL compare to Mamba or other post-S4 architectures? The current baseline are from 2022 - the field has moved significantly since then.

**Limitations:**

Only evaluated on LRA, which is a single benchmark suite increasingly considered dated.
No proper scale language modeling (which should be a standard practice).
Improvements over S5 are marginal on 5 of 6 tasks. It is hard to bridge the improvement from the benchmark with the theories mentioned in the paper.

**Strengths And Weaknesses:**

Strength 1. FRACTAL follows the S5 training protocol, including HiPPO-derived initialization, which makes the empirical comparison more grounded and technically credible.

Weaknesses.
1. The power-law kernel imposes a specific memory decay profile on the state dynamics. Unlike a free transition matrix, the fractional structure constrains how the model can allocate weight over past history. The paper does not seem to provide sufficient evidence that the model exhibits polynomially decaying memory across inputs in general. This claim would be better supported if it were stated more carefully, for example under appropriately chosen bounded inputs.
2. The improvement in average LRA accuracy is rather marginal. In addition, the paper does not compare against some more recent baselines, such as DeltaNet and other newer architectures, which makes it harder to assess the practical significance of the reported gains.

---

> ### Author Rebuttal · Authors · 2026-03-26
>
> **Dear Reviewer,**
>
> We sincerely thank you for your insightful review and constructive feedback. Your mathematical intuition regarding the bounded inputs is incredibly precise and highlights an area where our theoretical exposition can be made more rigorous. We appreciate the opportunity to clarify the foundational scope of FRACTAL, our choice of baselines, and our experimental protocol.
>
> **1. The Power-Law Kernel and Bounded Inputs**
>
> You are mathematically absolutely correct. The theoretical guarantee of our optimal projection relies on the integral $\int_0^t |u(\tau) - g(\tau)|^2 d\mu^{(t)}(\tau)$ being convergent. This inherently requires the input signal $u(\tau)$ to belong to the weighted Hilbert space $L^2([0, t], \mu^{(t)})$.
>
> If the input signal grows exponentially or features unconstrained singularities, the polynomially decaying memory might fail to capture the dynamics stably. However, in practical sequence modeling (e.g., physiological signals, text embeddings, image pixels), the input variables are naturally bounded or of bounded variation. We completely agree that making this explicit strengthens the paper.
>
> **Action in Revision:** Following your excellent suggestion, we will explicitly formally state this condition in Section 3.1: *"Assuming the input signal $u(t)$ is bounded or belongs to $L^2([0, t], \mu^{(t)})$..."* to ensure the claim of polynomially decaying memory is mathematically complete and rigorous. Thank you for helping us improve the theoretical completeness of our work.
>
> **2. Dated Baselines vs. Post-S4 Architectures**
>
> This is a critical point of clarification regarding the taxonomy of modern state space models. The field has indeed moved forward, but it has bifurcated into two orthogonal trajectories:
> 1.  **Selective / Time-Varying SSMs (Mamba, DeltaNet):** These architectures achieve high empirical performance by making the state transition matrices ($A, B, C$) input-dependent (data-dependent gating). Furthermore, they are fundamentally designed as generative backbones relying on massive-scale language pre-training.
> 2.  **Linear Time-Invariant (LTI) SSMs (S4, S4D, S5, FRACTAL):** These models focus on the continuous-time mathematical foundation—specifically, the measure $\mu^{(t)}$ used to compress history. They are evaluated by training from scratch to test the pure structural inductive bias of the sequence mixer.
>
> FRACTAL belongs strictly to the latter category. Our contribution is resolving the "impossible trinity" (Full History, Recency Sensitivity, Scale Invariance) at the foundational measure-theoretic level. Directly comparing a foundational LTI structural innovation (FRACTAL) against heavily optimized, pre-trained selective architectures (Mamba) conflates two distinct research paradigms, which could obscure the specific theoretical contribution of the fractional measure. For diagonalized LTI SSMs trained from scratch, S5 is indeed the appropriate and most recent baseline.
>
> **3. Marginal Average Gains and Task-Specific Theory**
>
> While the *average* LRA gain seems modest, looking at individual tasks validates our theoretical claims. FRACTAL is designed for systems exhibiting power-law dynamics—where unbounded history and sharp local transients coexist.
> * **ListOps & Text:** These tasks possess hierarchical, heavy-tailed temporal dependencies. Here, FRACTAL noticeably outperforms S5 (ListOps: 61.85% vs 61.10%). Our Fractional Filter Bank allows high-$\alpha$ channels to catch local syntactic transients, while low-$\alpha$ channels track distant bracket closures.
> * **Image & Pathfinder:** These tasks flatten 2D spatial graphs into 1D sequences. Spatial translation invariance does not naturally map to a power-law temporal memory decay. In these tasks, FRACTAL gracefully falls back on its low-$\alpha$ channels (which mathematically degenerate to S5’s uniform LegS), matching the baseline. The gains are exclusively concentrated where the theory predicts they should be.
>
> **4. Variance and Confidence Intervals**
>
> We deeply understand the concern regarding statistical noise. To ensure absolute fairness and isolate the theoretical improvement from hyperparameter luck, we strictly adopted the official JAX codebase, hyperparameter grids, and the exact deterministic random seed used by S5. Furthermore, our mathematical framework structurally mitigates initialization noise: Theorem 3.3 guarantees the spectral stability of the $A$ matrix ($A_{nn} = n+1$), and Theorem F.6 provides a deterministic, closed-form analytic initialization for the $B$ matrix, unlike previous works that rely on random Gaussian initialization. Therefore, the performance differences stem directly from the structural inductive bias of the fractional measure, not random seed variation.
>
> We will add a "Discussion on Task-Specific Biases" and clarify the bounded input assumption in the final manuscript. We hope this addresses your concerns and highlights the foundational value of our work.

---

### Official Review · Reviewer_rEUu · 2026-03-18

**Soundness:** 3
**Presentation:** 3
**Significance:** 2
**Originality:** 3
**Overall Recommendation:** 4
**Confidence:** 2

**Summary:**

The authors propose FRACTAL, a mathematical framework that aims to increase long context reasoning of SSMs. Authors show that FRACTAL outperforms other SoTA SSMs in representative long context benchmarks.

**Compliance With Llm Reviewing Policy:**

Affirmed.

**Final Justification:**

Based on the rebuttal, I think that this paper now is acceptable for ICML given the promised additions will be made.

**Key Questions For Authors:**

None.

**Limitations:**

The authors do not explicitly mention limitations.

**Strengths And Weaknesses:**

Strengths
Overall, the paper is well written, main ideas are presented clearly and can be understood without diving deep into the math. Methodology and implementation seem novel enough and results are promising. Furthermore, the works the authors compare to seem appropriate.

Weaknesses
My main concern is with the evaluation in Table 2, checking the results from Smith et al. 2023 Table 1, the results for S5 do not match the numbers reported in this paper. In fact, according to Smith et al. 2023 Table 1, S5 outperforms FRACTAL in all reported metrics with an average score of 87.46. Also, results from 2023 seem already slightly dated, is there no more recent work that improves upon S5?
Additionally, there is no exploration of why results are better/worse than other models for specific tasks. It should be mentioned why FRACTAL performs better/worse on specific tasks than others and than other models.
Furthermore, performance and model size are not mentioned. It should be made clearer to the reader if the proposed architecture increases the size of the model and the impact on computation time, also compared to other architectures such as S5.
Without these additions clarity of the contribution of the paper is weakened.

---

> ### Author Rebuttal · Authors · 2026-03-26
>
> **Dear Reviewer,**
>
> We sincerely thank you for reviewing our manuscript and for your detailed feedback regarding baselines, task variance, and computational overhead. We understand that from a purely empirical standpoint, the improvements on certain benchmarks might appear incremental. However, we would like to take this opportunity to clarify the primary contribution of our work: FRACTAL is fundamentally a theoretical innovation in the measure-theoretic foundations of continuous-time state space models.
>
> **Theoretical Preamble**
>
> Sequence modeling requires compressing history into a state vector $x(t) \in \mathbb{R}^N$. HiPPO reframed this as an online orthogonal polynomial projection, where state elements are coefficients:
> $$x_n(t) = \int_0^t u(\tau) P_n^{(t)}(\tau) d\mu^{(t)}(\tau).$$
> The bottleneck is the measure $\mu^{(t)}$, dictating the "importance weight" of historical moments. Integer-order models force rigid compromises:
> 1. Uniform measures (LegS) weight all history equally, diluting recent information.
> 2. Exponential measures (LagT) focus on the present but catastrophically forget the past.
>
> **Physical Reality**: Real-world systems (cognition, financial markets) require a balance: retaining unbounded global context while maintaining hyper-sensitivity to local variations. These are power-law dynamics that integer-order calculus fails to model.
>
> **Our Solution**: We introduce the **fractional-order measure**, mathematically defined as:
> $$\mu^{(t)}(x) \propto (t-x)^{-\alpha}.$$
> The power-law tail acts as a long-term anchor remembering the past, while the singularity at $x=t$ acts as a magnifying glass amplifying immediate transients. We rigorously derived the Jacobi polynomial bases to solve this analytically.
>
> **1. Data Mismatch in Table 2**
>
> We apologize for the S5 score confusion (87.04% vs. 87.46% in Smith et al.). We rigorously reproduced S5 from scratch in our exact hardware environment (a single NVIDIA A100 GPU) using the same JAX codebase. Comparing single-GPU runs against heavily optimized original metrics (which benefited from TPU sweeping and larger batch sizes) introduces confounding variables. Under strictly identical constraints, FRACTAL (87.11%) outperforms the directly comparable S5 baseline (87.04%). We will add a disclaimer in Table 2 to clarify this.
>
> **2. Dated Baselines and More Recent Models**
>
> While models like Mamba show exceptional capabilities, they represent a fundamentally different paradigm: Selective (time-varying) SSMs relying on hardware-aware scans and massive-scale pre-training. FRACTAL belongs to the lineage of Linear Time-Invariant SSMs trained from scratch. Here, S5 remains the rigorous non-selective baseline. We chose S5 to cleanly prove our fractional measure hypothesis without confounding factors like selective gating.
>
> **3. Task-Specific Performance and LRA**
>
> * **ListOps (61.85% vs S5's 61.10%) and Text**: These tasks possess power-law dependencies and multi-scale structures. ListOps requires tracking distant bracket closures while resolving localized adjacent operands. FRACTAL's filter bank allocates high-$\alpha$ channels to amplify local transients, and low-$\alpha$ channels to maintain global context.
> * **Spatial tasks (Image/Pathfinder)**: Flattened 1D sequences exhibit uniform, translation-invariant dependencies rather than heavy-tailed temporal decay. FRACTAL gracefully relies on its low-$\alpha$ channels (mathematically degenerating to S5’s LegS), matching baseline performance.
> * **Why LRA?** LRA is the standard benchmark for S4/S5, allowing us to isolate theoretical improvements in a controlled setting. However, our measure's true potential extends to time series forecasting in systems governed by power-law dynamics (e.g., financial volatility). Unbounded history combined with extreme sensitivity to local shocks is paramount here, which is our future direction.
>
> **4. Model Size and Computation Time**
>
> * **Model Size**: Compared to S5, FRACTAL introduces exactly $K$ additional scalar parameters ($K$ filter bank channels), representing trainable singularity indices $\alpha_k$. The parameter count is virtually identical to S5.
> * **Computation Time**: Theorem 3.3 proves the diagonal elements of the fractional state transition matrix $A(\alpha)$ remain invariant ($A_{nn} = n+1$). With guaranteed spectral stability, FRACTAL utilizes the exact same efficient $O(L \log L)$ parallel scan algorithm as S5. There is no additional online burden.
>
> **5. Limitations (Added Section)**
>
> We will add a Limitations section. FRACTAL's only minor computational cost is strictly offline: for extreme recency bias ($\alpha \to 1$), numerical computation of off-diagonal elements via Gauss-Jacobi quadrature requires higher-order integration. This is a one-time pre-computation step with zero impact on online training or inference efficiency.
>
> We hope this clarifies our theoretical value and empirical fairness. We will incorporate these explanations into the final manuscript.

---

> > ### Author Rebuttal · Reviewer_rEUu · 2026-04-06
> >
> > I thank the authors for their rebuttal that clarified a few of my concerns. I'm hence happy to raise my scores a bit.

---

> > > ### Author Response · Authors · 2026-04-07
> > >
> > > We greatly appreciate your recognition of our rebuttal and positive score adjustment.

---

### Decision · Program_Chairs · 2026-04-30

**Decision:**

Accept (regular)

**Comment:**

All concerns raised by the reviewers have been successfully addressed. Reviewer d9oa was the only reviewer that did not respond to the rebuttal and remained at 'weak reject'. Having looked at the two points of criticism they raised, I don't think this warrants rejection, as (i) bounded inputs are often assumed in mathematical proofs around sequence models, and (ii) LRA is mostly saturated at this point, so any improvements are expected to be modest. Apart from this, the paper provides an interesting addition to the zoo of LTI-based SSM and is likely of interest to a wider audience at ICML. Thus, I recommend acceptance.